# A Source for the Continuous Generation of Pure and Quantifiable HONO Mixtures

Guillermo Villena[1] and Jörg Kleffmann[1]

[1] Department of Physical and Theoretical Chemistry, Faculty of Mathematics and Natural Sciences, University of Wuppertal, 42097 Wuppertal, Germany

*Correspondence to*: Jörg Kleffmann (kleffman@uni-wuppertal.de)

**Abstract**

A continuous source for the generation of pure HONO mixtures was developed and characterized, which is based on the Henry's law solubility of HONO in acidic aqueous solutions. With the help of a peristaltic pump diluted nitrite and sulfuric acid solutions are mixed in a temperature-controlled stripping coil, which is operated with pure nitrogen or synthetic air at gas flow rates of 0.5-2 L min$^{-1}$. Caused by the acidic conditions of the aqueous phase (pH $\approx$ 2.5), nitrite is almost completely converted into HONO, which partitions to the gas phase limited by its known solubility in water. The source shows a fast time response of ~2 min (0-90 %) at higher concentrations and an excellent long-time stability (2 $\sigma$ noise <1 %). The HONO emission of the source perfectly correlates with the nitrite concentration from the sub-ppb range up to 500 ppb. The rate of NO$_x$ formation increases quadratically with the HONO concentration from non-detectable values at atmospheric relevant HONO concentrations reaching a NO$_x$ content of 1.6 % at 500 ppb. A general equation based on Henry's law is developed, whereby the HONO concentration of the source can be calculated using measured experimental parameters, i.e. nitrite concentration, liquid flow rates, gas flow rate, pH of the solution and temperature of the stripping coil. In the equation, the known Henry's law constant of HONO in sulfuric acid solutions is used. For the calculation of the effective Henry's law constant, the acid dissociation equilibrium of HONO/nitrite is used as a variable to adjust the theoretical HONO concentration to the measured values. From the average of all experimental data the equilibrium of HONO/nitrite is well described by p$K_a$ = 1021.53/T - 0.449. The p$K_a$ of 3.0±0.1 (1$\sigma$) at 25 °C is in good agreement with the range of 2.8-3.28 published in former studies. A standard deviation between all measured and theoretical HONO concentrations of only ±3.8 % was observed and a conservative upper limit accuracy of the HONO concentration of better 10 % is estimated. Thus, for the first time, a stable HONO source is developed, which can be used for the absolute calibration of HONO instruments.

# 1    Introduction

Nitrous acid (HONO) is an important trace gas in the atmosphere, which represents a major source of the OH radical (Kleffmann et al., 2005; Acker et al., 2006; Ma et al., 2013; Michoud et al., 2014; Gu et al., 2020), the detergent of the
atmosphere. In addition, HONO and its reaction products show mutagenic and carcinogenic properties (Pitts et al., 1978; Kirchner and Hopkins, 1991), which is especially important for indoor conditions, for which much higher levels of up to 90 ppb compared to the ambient atmosphere have been observed (Večeřa and Dasgupta, 1994).

The sources of HONO in the atmosphere are controversially discussed (Kleffmann, 2007) and among others, photochemical heterogeneous reactions have been proposed to explain unexpected high HONO concentrations during daytime (Stemmler et
al., 2006; Zhou et al., 2011). However, the accuracy of the daytime HONO data was often doubted, for which spectroscopic instruments are often not sensitive enough, while wet-chemical instruments may suffer from chemical interferences (Kleffmann and Wiesen, 2008). Thus, the development of sensitive and selective HONO instruments is still of high interest, for which a stable and pure HONO source is very helpful. Such a HONO source should ideally be simple in operation, produce HONO mixtures with high purity and stability, cover a wide range of concentrations and should also produce predictable
HONO concentrations for the calibration of HONO instruments in the field, when other calibration methods are not available. Several attempts have been made in the past to produce HONO mixtures in the laboratory and in the field. In first studies, simply the equilibrium between nitrogen oxides (NO+NO$_2$) and water vapor was used to generate HONO (Wayne and Yost, 1951):

(1)        $NO + NO_2 + H_2O \rightleftharpoons 2\ HONO$ .

However, equilibrium (1) is never quantitative and the purity of HONO is typically lower than 50 %. Accordingly, other methods were developed, which are mainly based on different acid displacement reactions, for example using oxalic acid (Braman and de la Cantera, 1986):

(2)        $NaNO_2 + H_2C_2O_4 \longrightarrow HONO + NaHC_2O_4$,

sulfuric acid (Taira and Kanda, 1990):

(3)        $NaNO_2 + H_2SO_4 \longrightarrow HONO + NaHSO_4$,

or hydrochloric acid (Febo et al., 1995):

(4)        $NaNO_2 + HCl \longrightarrow HONO + NaCl$.

In addition to the acid displacement, Večeřa and Dasgupta (1991) developed a source, where the thermal decomposition of ammonium nitrite was used:

(5)        $NH_4NO_2 \longrightarrow HONO + NH_3$.

The purity of the latter source is however low, caused by the equimolar formation of HONO and NH$_3$. The different types of sources used in various studies were recently summarized in Table 1 of Gingerysty and Osthoff (2020).

In most recent studies, the acid displacement using HCl was applied (Ren et al., 2010; Reed et al., 2016; Gingerysty and Osthoff, 2020; Lao et al., 2020), which are all based on the pioneering study of Febo et al. (1995). Gaseous HCl is produced by any kind of permeation source and after humidification reacts heterogeneously with solid sodium nitrite. After careful optimization, high purity and a large concentration range from ppb to ppm levels can be obtained by the source (Febo et al., 1995). However, when using HCl under dry conditions or at low relative humidity, there is always a risk of undesired formation of ClNO. In addition, unreacted HCl may occur, when the sodium nitrite reactor is not working properly (Gingerysty and Osthoff, 2020). For this reason, we used this source only for a short period in our laboratory (Brust et al., 2000) and later switched to the experimentally much simpler source from Taira and Kanda (1990), for which the above-mentioned by-product formation is completely absent, when using the non-volatile $H_2SO_4$. In the source, diluted nitrite and $H_2SO_4$ solutions are mixed in a temperature controlled bubbler-type reactor, while flushed with synthetic air. The HONO concentration of the source can be adjusted simply by changing the concentration of the aqueous nitrite solution. In contrast, for the source by Febo et al. (1995), variations of the temperature and the concentration of the liquid HCl in the permeation source are required, which is more complicated and time-consuming. Finally, the source of Taira and Kanda (1990) was optimized by using a temperature-controlled stripping-coil reactor, leading to even higher simplicity and stability, faster time response and to lower $NO_x$-formation (Kleffmann et al., 2004). The source was later commercialized (QUMA Elektronik & Analytik GmbH) and its prototype is in regular use in our laboratory and for the characterization of different HONO instruments (e.g. Jurkat et al., 2011; Ródenas et al., 2013). However, in addition to the very few information given in Kleffmann et al. (2004), the source was never explained in detail. Caused by the increasing general interest in the chemistry of HONO and in the development of simple HONO sources during the last years (e.g. Gingerysty and Osthoff, 2020; Lao et al., 2020), the aim of the present study was the characterization of the source in more detail and also to apply Henry's law for its theoretical quantification to be used for the stand-alone calibration of HONO instruments.

## 2    Experimental

### 2.1    HONO Source

The set-up of the source is shown in Figure 1. Nitrogen ($N_2$) from evaporation of 99.999 % pure liquid nitrogen (alternatively also synthetic air can be used) controlled by a 2 L min$^{-1}$ flow controller (Bronckhorst) is passed through a temperature controller stripping coil (2.4 mm i.d. glass tube, 25 turns, 20 mm turn diameter) with gas flow rates in the range 0.5-2 L min$^{-1}$. The temperature of the stripping coil is controlled in the range 5-20 °C by a compact liquid thermostat (QUMA, Peltier type) and a thermoelement, which was calibrated by a certified Hg-thermometer. The maximum temperature of the stripping coil is limited to a few °C below room temperature. If higher temperatures are used, water will condensate in the PFA (perfluoroalkoxy alkanes) lines (4 mm i.d.). For most experiments, the source was operated at the nominal instrument set-point of 15 °C, which represents a real calibrated temperature of 15.9 °C.

In the stripping coil, the gas phase comes into contact with a mixture of diluted nitrite and $H_2SO_4$ solutions. Except for the study of the pH dependence, a 3.6 mM $H_2SO_4$ solution was used, for which 2 mL of a 1/10 diluted $H_2SO_4$ (Aldrich p.a., 95-98 %) was diluted to 1 L with pure water. Diluted nitrite solutions (0.001-10 mg $L^{-1}$) were made from a 1000 mg $L^{-1}$ standard stock solution (Merck Titrisol) by dilution with volumetric pipettes and flasks. If nitrite concentrations $\geq$0.1 mg $L^{-1}$ are used, these solutions are stable for weeks, when stored in the dark. For lower concentrations daily preparation is recommended. The pH of the combined solutions of ~2.5 collected from the waste channel of the instrument was measured by a calibrated pH-meter (HANNA instruments, HI8314 membrane pH meter). The liquid flow rates of the two reagent solutions and of the waste are controlled by an 8-channel peristaltic pump (Ismatec) using each 0.51 mm i.d. peristaltic pump tubing for the nitrite and $H_2SO_4$ solutions and a 1.14 mm i.d. tubing for the waste (Ismaprene, PharMed®, three stopper, lifetime 2×1000 h), respectively. The peristaltic pump can be operated from 8-40 rpm and the liquid flow rates were measured by the time to fill 5 ml volumetric flasks. The liquid flow rate of a peristaltic pump is typically decreasing by ~20 % during the lifetime of the tubes and should be measured when the theoretical HONO concentrations are calculated. All components described above including plastic bottles for the reagents (2×0.5 L) and the waste (1 L) are installed in a 19'' rack housing (see Figure 1).

## 2.2 Other Analytical Equipment

For the measurement of HONO and $NO_x$ a chemiluminescence $NO_x$ monitor (Eco-Physics, nCLD SL899) with molybdenum converter was used. The instrument has a detection limit of ~30 ppt at a sample flow rate of 1 L $min^{-1}$ and was daily calibrated by a certified NO calibration mixture (410 ppb NO, Messer) with an accuracy of 3 %. While the instrument is very selective for the detection of NO, all other reactive nitrogen species ($NO_y$), i.e. for the present study $NO_2$ and HONO, are quantitatively measured in the $NO_x$ channel of the instrument (Villena et al., 2012). Quantitative $NO_2$ conversion was assured by the replacement of the molybdenum converter during the annual maintenance of the instrument just a week before the experiments and was also verified with an $O_3$-titration unit (Anysco GPT). Quantitative conversion of HONO to NO was demonstrated by comparison with a HONO-LOPAP instrument (Heland et al., 2001; Kleffmann et al., 2002). To confirm that HONO decomposes to NO and $NO_2$ at equimolar ratio according to the back reaction (1) also an $NO_2$-LOPAP instrument (Villena et al., 2011) was used during the concentration dependency.

The HONO mixture from the source was guided by PFA-lines (4 mm i.d.) to the different instruments. The excess flow passed a humidity and temperature sensor (HYTELOG-USB, Hygrosens Instruments GmbH, accuracy ±2% RH) to measure also the dewpoint of the gas phase. Caused by this set-up, the lower gas flow rate of the HONO source applied in the present study was limited by the flow rate of the chemiluminescence instrument (1 L $min^{-1}$).

## 3 Results

At HONO mixing ratios higher than 100 ppb significant formation of $NO_x$ was observed (see below section 3.2), which is caused by the decomposition of HONO, see back reaction (1). In agreement with the stoichiometry of the reaction, the $NO_2$

concentration measured by the NO$_2$-LOPAP was found to be similar to the NO concentration measured by the chemiluminescence instrument. Thus, for simplicity, the NO$_x$ level of the HONO source was calculated by doubling the measured NO signal. The HONO concentration measured by the chemiluminescence monitor was then determined by the difference between [NO$_y$] and 2×[NO]. The calculated HONO concentrations agreed well with those from the HONO-LOPAP in between the experimental errors of both instruments. During a concentration dependency experiment in the range 0-100 ppb

(see section 3.2) both instruments showed an excellent linear correlation (R$^2$ = 0.9995) and agreement (slope LOPAP against ECO: 0.972). Since the accuracy of the ECO-Physics instrument is slightly higher (3 %) than that of the HONO-LOPAP (5 %) the chemiluminescence HONO data were further used for the characterization of the HONO source. Only for one low concentration experiment (see section 3.6) the HONO-LOPAP data is shown.

### 3.1    pH Dependence

The effective solubility of HONO in aqueous solution is limited by different processes. First, the solubility of undissociated HONO is described by Henry's law:

(I)        $K_H = \dfrac{c_{lq.}}{p_g}$,

where the liquid concentration $c_{lq.}$ is given in (mol L$^{-1}$) and the gas phase partial pressure of HONO $p_g$ in (atm). In former studies consistent values of the Henry's law constant $K_H$ (mol L$^{-1}$ atm$^{-1}$) were observed (e.g. Park and Lee, 1988; Becker et al.,

1996). Second, since H$_2$SO$_4$ solutions were used in the present study, the salting out effect of the weak acid HONO by the strong acid H$_2$SO$_4$ has to be considered. Thus, for the HONO/H$_2$SO$_4$ system the solubility data from our former study Becker et al. (1996) was used, which agreed well with the results from Park and Lee (1988) for pure water. Furthermore, considering that only a moderate acidity of the reaction mixture was finally used (pH ≈ 2.5, see below), the added nitrite is still not completely converted into undissociated HONO in the aqueous phase, caused by the pH-dependent acid dissociation

equilibrium:

(6)        HONO        $\rightleftharpoons$        NO$_2^-$ + H$^+$.

To quantitatively describe the equilibrium (6), p$K_a$ values in the range 2.8-3.26 have been published in former studies (Park and Lee, 1988; Riordan et al., 2005; da Silva et al., 2006). Considering this equilibrium, the effective Henry's law constant $K_{H,eff.}$ (mol L$^{-1}$ atm$^{-1}$) is given by:

(II)        $K_{H,eff.} = K_H \left(1 + \dfrac{K_a}{c_{H+}}\right)$,

where the H$^+$ concentration $c_{H+}$ (mol L$^{-1}$) is calculated from the measured pH of the combined solutions (waste). Since the acid dissociation constant has a significant uncertainty, we used the p$K_a$ as unknown variable to adjust the theoretical HONO concentrations (see equation (III), section 4.2) to the experimental values. When using all experimental data at pH >2.4 and excluding the temperature dependence, an average p$K_a$ = 3.09±0.12 (1σ) was determined at the typical used temperature of

15.9 °C. When the temperature dependence was also considered (see section 3.5) the equilibrium is well described by $pK_a$ = 1021.53/T - 0.449 in the experimental temperature range (5.4-18.6 °C). The resulting $pK_a$ of 3.0±0.1 (1σ) at 25 °C is in good agreement with the range of 2.8-3.28 published in former studies (Park and Lee, 1988; Riordan et al., 2005; da Silva et al., 2006).

When the pH of the solution was varied, an excellent agreement between theoretical (see section 4.2) and experimental HONO concentrations was observed only for pH values >2.4. For lower pH significantly lower HONO concentrations were observed in comparison with the values calculated with equation (III) (see Figure 2). Reasons for this discrepancy are still unclear and could not be explained by the increasing rate of $NO_x$ formation with decreasing pH, as even the measured $NO_y$ concentration was lower than the theoretical HONO concentration (see Figure 2). One potential problem could be the pH measurements, which showed excellent agreement between measured and theoretical values only for pH >2. In contrast, at higher acidity measured pH-values were significantly higher than theoretically expected, which is a known problem when using pH-electrodes (Bates, 1973). Thus, the theoretical and not the experimental pH-values were used for pH <2 in Figure 2. For calculation of the theoretical pH, the acid concentration was used and reasonable quantitative dissociation of the strong $H_2SO_4$ was assumed. Furthermore, according to the study by Riordan et al. (2005) a second equilibrium between $HONO_{aq.}$ and $H_2ONO^+_{aq.}$ at pH <2 could lead to a higher effective solubility compared to undissociated HONO. However, in our former study Becker et al. (1996), the equilibrium between HONO and $NO^+$ was observed at much higher acidity around 55 wt%, which was in good agreement with a former study by Seel and Winkler (1960). Thus, any increasing effective solubility by formation of $H_2ONO^+/NO^+$ should not be of importance in the pH range 0-2. Finally, liquid phase diffusion in the stripping coil could limit the experimental HONO emission with increasing viscosity of the $H_2SO_4$ solutions at concentrations up to 2.7 wt% used, leading to lower HONO concentrations than theoretically expected from the thermodynamic equilibrium. In contrast, in our study by Becker et al. (1996) on which equation (III) is based, a bubbler set-up was used under thermodynamic equilibrium.

Since our focus was not the exact study of the system N(III) in sulfuric acid solutions, but the development of a HONO source, we not further investigated this issue in detail, but simply limited the acidity to pH ≥2.4. For these conditions, the pH-measurements were accurate, the theoretical and experimental HONO concentrations agreed well, a potential equilibrium of HONO and $H_2ONO^+$ is still not significant (Riordan et al., 2005) and the unwanted $NO_x$ formation is still small (see Figure 2). The pH should however also not be too high (i.e. >3), since the uncertainties in the theoretical HONO concentrations increase at higher pH, caused by the incomplete conversion of the added nitrite to HONO, see equilibrium (6). Thus, for the present HONO source an acidity of the reaction mixture of pH ≈ 2.5 is recommended.

### 3.2 Concentration Dependence

The dependence of the added nitrite concentration on the HONO mixing ratio was studied in two different experiments and by two different operators of the source. The experimental data from one experiment is shown in Figure 3, for which the nitrite concentration was varied by several orders of magnitude (0.002 – 10 mg $L^{-1}$). The source showed a very fast time response of

~2 min (0-90 %) for HONO concentrations >20 ppb increasing to 6-7 min at lower concentrations at a liquid pump speed of 20 rpm. The increasing time response at low HONO levels is explained by adsorption/desorption of HONO on the surfaces behind the HONO source, which gets less important with increasing HONO levels, leading to faster saturation of the surfaces. From the experiment shown in Figure 3, we conclude that most of this adsorption/desorption took place on the surfaces of the chemiluminescence instrument used (inlet particle filter, stainless-steel lines) and not on the PFA transfer lines. At 16:09 the HONO source was switched from reagents to pure water, for which the HONO emissions should quickly decrease to zero. However, after a first fast decrease of the HONO concentration there was a significant tailing of the signal. Here the slope of the decreasing signal did not change when the HONO source was replaced by pure nitrogen at 16:59. This can only be explained when the tailing is caused by desorption of HONO from the surfaces of the chemiluminescence instrument, as all other PFA surfaces were removed. This conclusion is also confirmed by the LOPAP data, for which a time response of only 4 min was observed at low mixing ratios in the range 0.05 – 0.5 ppb (see section 3.6 and Figure 8). Since this time response is similar to the one of the LOPAP instrument under the experimental conditions applied, the time response of the HONO source will be ≤4 min, in agreement with the high concentration data. The proposed adsorption of HONO mainly on the metal surfaces of the chemiluminescence instrument at low HONO levels is also in agreement with our experience with pure HONO mixtures, for which adsorption losses in PFA transfer lines of up to 20 m length were found to be insignificant.

When all HONO data from the two experiments were plotted against the nitrite concentration an excellent linear correlation ($R^2$ = 0.99996) was observed. Similar slopes were observed when using all HONO data up to 500 ppb (see Figure 4 A) compared to mixing ratios only up to 10 ppb (see Figure 4 B). At typical atmospheric HONO mixing ratios <20 ppb no $NO_x$ formation was observed in between the experimental errors. Only at higher HONO levels, $NO_x$ quadratically increased with the HONO concentration (see Figure 4 A), in agreement with the expected second order kinetic behaviour of the back reaction (1). But even at 500 ppb HONO, a $NO_x$ content of only 1.6 % was observed, which is significantly lower than in the original bubbler set-up of Taira and Kanda (1990). Thus, the HONO source can be operated for mixing ratios up to 500 ppb and up to 20 ppb with purities of >98 % and >99.8 %, respectively. In addition, the excellent agreement of the HONO and $NO_x$ data for the two completely independent experiments performed by two different operators including different $NO_x$ calibration and dilution of the nitrite stock solution (see Figure 4) demonstrates the high precision and reproducibility of the HONO source (see also section 3.6).

### 3.3    Gas Flow Rate Dependence

The used stripping coil can be operated at gas flow rates in the range 0.5-2 L min$^{-1}$. However, caused by the sample flow rate of the chemiluminescence instrument of 1 L min$^{-1}$, only gas flow rates in the range 1-2 L min$^{-1}$ were investigated. The HONO concentration was decreasing with the gas flow rate (see Figure 5), which can be explained by the increasing dilution of the formed HONO in the increasing gas volume.

Furthermore, a decreasing $NO_x$ content with increasing gas flow rate was observed, which is explained by the decreasing reaction time of HONO in the stripping coil and the heterogeneous decomposition of HONO (back reaction (1), second order

kinetics). Thus, for increasing the purity of the source, it should be operated at gas flow rates >1.5 L min$^{-1}$. If lower flow rates are necessary, the HONO levels should be reduced to mixing ratios <10 ppb, caused by the quadratic dependence of the NO$_x$ formation with the HONO mixing ratio, see section 3.2. The time response of the source was found to be independent of the gas flow rate.

### 3.4  Liquid Flow Dependence

Since the amount of nitrite pumped into the stripping coil is directly proportional to the liquid flow rate, the speed of the liquid peristaltic pump was varied in the range 10-40 rpm. As expected, the HONO mixing ratios increased with the liquid flow rate (see Figure 6). However, the increase was found to be non-linear, since at higher liquid flow rates the liquid volume in the stripping coil is also increasing. Thus, an increasing content of nitrite/HONO is remaining in the liquid phase according to

Henry's law.

No significant impact of the NO$_x$ formation on the liquid flow was observed (see Figure 6). This could be explained by the heterogeneous back reaction (1) that only occurs on the surface of the stripping coil, which is not affected by the liquid flow rate. In contrast, if the decomposition of HONO proceeded via a liquid phase reaction, the NO$_x$ content should increase with the liquid flow rate. As expected, the time response of the source was observed to decrease with the liquid flow rate, which

can be explained by faster liquid exchange in the stripping coil.

### 3.5  Temperature Dependence

Since the solubility of HONO is temperature dependent (Park and Lee, 1988), also the temperature of the stripping coil was varied in the range 5.4-18.6 °C. The maximum temperature of the stripping coil is limited to a few degrees below the laboratory temperature, to exclude condensation of water on surfaces behind the stripping coil. Here, strong fluctuation of the HONO

concentration was observed when the source was operated close to the dewpoint of water. The lower temperature limit is given by the power of the Peltier cooler of the source. Besides the HONO levels, also the absolute humidity of the gas phase can be varied by the temperature of the stripping coil (see Clausius Clapeyron).

As expected from the known solubility data (Park and Lee, 1988; Becker et al., 1996), the HONO concentration of the source was found to increase with the temperature (see Figure 7). In contrast, the NO$_x$ content of the source was slightly decreasing

with the temperature, which was not caused by decreasing absolute NO$_x$ concentrations, but by the increasing HONO levels. Similar to the other experiments, the theoretical HONO concentrations calculated by equation (III) (see section 4.2) were adjusted to the measured values by varying the p$K_a$ for the acid dissociation equilibrium (6), leading to a decreasing p$K_a$ with increasing temperature; for the quantitative description, see section 3.1.

### 3.6  Stability Test

To test the long-term stability and precision, the source was operated overnight at constant experimental conditions at a low liquid pump speed of 10 rpm leading to reduced liquid consumption and a time response (0-90 %) of the source of ~4 min. At

high HONO mixing ratios of ~33 ppb the precision of the source was <1 % ($2\sigma$ noise: 0.76 %), see Figure 8. This high precision is similar to the precision of the chemiluminescence instrument used to quantify the HONO source. Thus, the given precision error of the source is even an upper limit. When a much lower mixing ratio of ~0.5 ppb was used, almost the same precision of 1.1 % was obtained (see Figure 8). However, caused by the lower HONO level applied, the much more sensitive LOPAP technique was used for quantification. The LOPAP had a detection limit of 2 ppt and a precision of 1.0 % for the experimental conditions applied, explaining the slightly lower precision of the data. In good agreement with the results shown in section 3.2, an excellent linear correlation of the HONO mixing ratio with the nitrite concentration was observed ($R^2$ = 0.9997) also in the very low concentration range of 0.05-0.5 ppb (see Figure 8). The two experiments show that the precision of the HONO source is not depending on the HONO concentration in agreement with the used concept of the source. The variability of the HONO emission is only depending on the mixing of the two reagents at the inlet of the stripping coil (see Figure 1) and the stabilities of the gas and liquid flows. These parameters will not change with the nitrite concentration.

With the reduced liquid flow rate and using the internal liquid containers of the instrument, the source can be continuously operated for ~43 hours. If longer operation is necessary 5 L (nitrite, $H_2SO_4$) and 10 L (waste) liquid bags could be used. However, for such long periods the liquid flow rate of a peristaltic pump is typically slightly decreasing with time, leading to some expected drift of the HONO source. In this case, the liquid flow rates (nitrite, waste) should be regularly measured and included in the calculation of the theoretical HONO concentration, see section 4.2.

## 4    Discussion

### 4.1    General Considerations

In the present study a new HONO source was developed and characterized. In contrast to most recent studies (Ren et al., 2010; Reed et al., 2016; Gingerysty and Osthoff, 2020; Lao et al., 2020), HONO is produced by the reaction of nitrite and $H_2SO_4$ in the liquid phase. In a stripping coil reactor HONO partitions to the gas phase according to its known moderate solubility in acidic solutions. The advantage of using $H_2SO_4$ is the missing formation of unwanted by-products like HCl and ClNO as observed in HONO sources using the volatile HCl, which are based on the original concept of Febo et al. (1995). While these sources can be carefully optimized for low by-product formation (Febo et al., 1995; Gingerysty and Oshoff, 2020; Lao et al., 2020), the use of the non-volatile $H_2SO_4$ completely excludes the still existing risk of any malfunction of the source. For example, when the Febo et al. source was used in our laboratory two decades ago (Brust et al., 2000), we often had problems related with the homogeneous mixing of the solid $NaNO_2$ and with the use of low relative humidity, leading to lower purity of HONO.

Further advantages of the new HONO source are:

- The high time response (0-90%) of 1.5-7 min depending on the liquid flow rates and the HONO concentration levels used. For a typical pump speed of 20 rpm and HONO levels >20 ppb a time response of 2 min was observed. In contrast, the sources of Taira and Kanda (1990) and Febo et al. (1995) show much longer time responses. Even when the original source

from Febo et al. (1995) was optimized by using a modified HCl permeation source and a solid NaNO$_2$ coated flow tube reactor, HONO stabilisation times of two hours were observed, after the HCl permeation source was running constantly (Lao et al., 2020). Furthermore, for the HCl permeation source stabilization times of a week and more were necessary, still leading to some unwanted peaks of the permeation rate after this long stabilization time (Lao et al., 2020). In contrast, the present HONO source can be started under water, which can additionally be used to zero any HONO instrument and after 1 h stabilisation of the liquid flow rate of the peristaltic pump stable HONO concentrations are obtained in a few minutes, when water is exchanged by the nitrite and H$_2$SO$_4$ reagents.

- The wide mixing ratio range of 0.05-500 ppb, simply by changing the nitrite concentration. Variable HONO mixing ratios can be important in laboratory studies, but are also necessary when HONO instruments with non-linear response are calibrated (Jurkat et al., 2011). A very linear correlation between HONO and nitrite is observed. Different concentrated nitrite solutions can be easily made with high accuracy by dilution of a stock solution using volumetric pipettes and flasks. In contrast, for the source of Febo et al. (1995), different concentrated HCl solutions and HCl temperatures have to be used to adjust the output of the source, which is much more difficult and time-consuming. Dilution of the source by synthetic air and using flow controllers is not recommended, caused by the decreasing precision of the source and the resulting variable humidity. The latter can be a problem, when a humidity dependent HONO instrument is characterized, e.g. when the CIMS technique is used (Jurkat et al., 2011). In contrast, for the present source, variable HONO concentrations are obtained for constant humidity.

- The high purity of the source. For HONO mixing ratios <20 ppb no NO$_x$ formation and a purity of >99.8 % is observed. And even at high HONO levels of 500 ppb the purity of the source is >98 %. In contrast, for other recent HONO sources only purities >95 % and >90% were observed, respectively (Gingerysty and Osthoff, 2020; Lao et al., 2020).

- The high long-term stability and precision of the source. After stabilisation, an upper limit 2σ precision of the source of 0.76 % was observed, which is similar to the precision error of the used NO$_x$ instrument.

- The wide variability of the experimental conditions. Here, the liquid and gas flow rates and the temperature can be varied, leading for example, to variable humidity, time response and reagent consumption of the source.

- The predictability of the HONO output. The absolute HONO concentration can be calculated based on Henry's law with high accuracy (see next section 4.2), which offers for the first time the possibility of using a HONO source for the absolute calibration of HONO instruments.

## 4.2    Theoretical Calculation of the HONO Concentration

When the acidity of the reaction mixture was fixed to pH ≈ 2.5, for which the effective solubility of HONO is well described by the Henry's law solubility and the acid dissociation equilibrium (6), see section 3.1, the experimental HONO mixing ratios of the source are calculated by equation (III):

(III)  $[HONO]_{theo.} = \dfrac{c_{NO_2^-} \cdot \phi_{NO_2^-} \cdot N_A}{M_{NO_2^-} \cdot \left(N_A \cdot \phi_{waste} \cdot K_{H_{eff}} \cdot p + \frac{N^{p,T}}{V} \cdot \phi_g^{p,T,r.h.}\right)} \cdot 10^{-9}$ (ppb),

where $c_{NO_2^-}$ represents the nitrite concentration (g L$^{-1}$), $\phi_{NO_2^-}$ the nitrite liquid flow rate (L min$^{-1}$), $N_A$ the Avogadro constant (6.02214·10$^{23}$ molecules mol$^{-1}$), $M_{NO_2^-}$ the molar mass of nitrite (46.005 g mol$^{-1}$), $\phi_{waste}$ the liquid flow rate of the waste (L min$^{-1}$), $K_{H_{eff}}$ the effective Henry's law constant (mol L$^{-1}$ atm$^{-1}$, see section 3.1), $p$ the ambient pressure (atm), $\frac{N^{p,T}}{V}$ the number density of the gas phase according to the ideal gas law at ambient pressure and temperature of the stripping coil (molecules

315 cm$^{-3}$) and $\phi_g^{p,T,r.h.}$ the gas flow rate at ambient pressure, temperature of the stripping coil and considering the evaporation of water at the temperature of the stripping coil (cm$^3$ min$^{-1}$). $\phi_g^{p,T,r.h.}$ is calculated by:

  (IV)  $\phi_g^{p,T,r.h.} = \phi_g^{\emptyset,dry} \cdot \dfrac{T^{exp.}}{T^\emptyset} \cdot \dfrac{p^\emptyset}{p^{exp.}} \cdot \dfrac{p^{exp.}}{(p^{exp.} - p^{water})}$,

where $\phi_g^{\emptyset,dry}$ is the gas flow rate (cm$^3$ min$^{-1}$) of the calibrated flow controller at dry standard conditions ($T^\emptyset$ = 298.15 K, $p^\emptyset$ = 1 atm) and $T^{exp.}$ and $p^{exp.}$ the experimental temperature (K) and pressure (atm), respectively. To calculated the water vapor

pressure $p^{water}$ (atm) the Magnus equation was used (Alduchov and Eskridge, 1996):

  (V)  $p^{water} = 6.1094 \cdot exp\left(\dfrac{17.625 \cdot t(°C)}{243.04 \quad t(°C)}\right) \cdot \dfrac{1}{1013.25}$ .

The measured liquid flow rate of the waste $\phi_{waste}$ was in excellent agreement with the sum of the flow rates of nitrite and H$_2$SO$_4$ considering the theoretical water evaporation in the stripping coil, confirming an almost complete saturation of the gas phase by water vapor at the temperature of the stripping coil. The liquid flow rates were measured by the time taken to fill up

5 ml volumetric flasks.

When all experimental data described in the previous sections, excluding the pH dependence (see section 3.1), were compared to the theoretical HONO concentrations, a weighted average ratio [HONO]$_{exp.}$/[HONO]$_{theo.}$ of 0.996 was observed. This excellent agreement is trivial, since the applied p$K_a$ was derived from the average of all individual adjusted values for fitting the theoretical HONO mixing ratios to the measured values, see section 3.1. However, more importantly, the average 1σ

standard deviation of all [HONO]$_{exp.}$/[HONO]$_{theo.}$ ratios was only 3.8 %, despite the high variability of the experimental conditions applied. When the less precise data at [HONO] <5 ppb were excluded, the average 1σ standard deviation of the [HONO]$_{exp.}$/[HONO]$_{theo.}$ ratios was even only 1.7 %, showing the very high precision of the source. The lower precision of the data at HONO mixing ratios <5 ppb is not caused by the precision of the HONO source, but by the lower precision of the chemiluminescence instrument used. This is confirmed by the low concentration data shown in Figure 8 determined by the

much more sensitive LOPAP technique, for which an average deviation between experimental and theoretical HONO mixing ratios of only 1.2±1.6 % was observed in the range 0.05-0.5 ppb.

Besides this high precision, also a high accuracy of better than 10 % is estimated for the HONO source, which is mainly based on the accuracy of the used chemiluminescence monitor (3 %), the precision error of all data for HONO concentrations >5 ppb

(1.7 %) and the errors in the different variables used in equation (III). For the liquid nitrite concentration an accuracy of typically 3-4 % is obtained, considering the accuracy of the nitrite stock solution of 1000 mg L$^{-1}$ (1 %) and those of the used volumetric pipettes and flasks. An accuracy of 2 % is estimated for the gas flow rate, since the flow controller was used at 50-100 % of its nominal flow rate. In addition, it was calibrated during the experiments with a soap bubble flow tube, for which temperature, pressure and water evaporation were carefully considered. Accuracies of the liquid flow rates of 1 % are estimated. The error of the theoretical HONO concentration induced by the pH-measurements is estimated to 1.5 %. And finally, the error introduced by the uncertainty of the stripping coil temperature of ±0.5 °C is 1 %. By error propagation a combined accuracy of ±5.8 % is obtained.

Because of the accuracy of better than 10 %, the HONO source can be used as a stand-alone device to absolutely calibrate HONO instruments, for which no simple calibration is available, e.g. mass spectrometers. In contrast, instruments which measure HONO after sampling in a liquid phase (HPLC, WEDD, LOPAP, etc.) can typically be calibrated by using liquid nitrite standards. However, even these instruments need careful characterization, for example of the sampling and detection efficiencies, where a pure HONO source would be also very helpful. And finally, the HONO source can be used to study the chemistry of HONO in the laboratory.

## 5    Conclusion

In the present study, a HONO source is developed and characterized, which is based on the effective Henry's law solubility of HONO in water. Diluted nitrite and sulfuric acid solutions are mixed in a temperature-controlled stripping coil, which is operated with pure nitrogen or synthetic air at gas flow rates of 0.5-2 L min$^{-1}$. Under the acidic conditions of the combined reaction mixture (pH ≈ 2.5), nitrite is almost completely converted into HONO, which partitions to the gas phase. The known Henry's law constant of HONO in H$_2$SO$_4$ is used for the calculation of the effective solubility. In addition, in the present study the acid dissociation equilibrium is described by p$K_a$ = 1021.53/T - 0.449. The p$K_a$ of 3.0±0.1 (1σ) at 25 °C is in good agreement with the range of 2.8-3.28 published in former studies. The source shows a fast time response of ~2 min (0-90 %) at higher concentrations and an excellent long-time stability (2 σ noise 0.76 %). The HONO emission of the source perfectly correlates with the nitrite concentration from the sub-ppb range up to 500 ppb. The rate of NO$_x$ formation increases quadratically with the HONO concentration from non-detectable values at atmospheric relevant HONO concentrations reaching a NO$_x$ content of 1.6 % at 500 ppb. A general equation based on the effective Henry's law solubility is developed, by which the HONO concentration of the source can be calculated using measured experimental parameters, i.e. nitrite concentration, liquid flow rates, gas flow rate, pH of the solution and temperature of the stripping coil. An average deviation between measured and theoretical HONO concentration of only ±3.8 % is observed and a conservative accuracy of the calculated HONO concentration of better 10 % is estimated. Thus, for the first time, a HONO source is developed, which can be used for the absolute calibration of HONO instruments.

## 6    Acknowledgement

We would like to thank the three anonymous referees for their comments, which helped to improve our manuscript.

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

**Figure Captions:**

**Figure 1: Picture of the HONO source (QUMA) and its experimental set-up.**

**Figure 2: Dependence of the HONO mixing ratio and the ratio $NO_x$/HONO on the pH of the $NO_2^-$/$H_2SO_4$ reaction mixture ($NO_2^-$ = 1 mg L$^{-1}$; T = 15.9 °C, $\phi_{g,dry}^{\emptyset}$.= 1570 cm$^3$ min$^{-1}$, 20 rpm). In addition, the theoretical HONO mixing ratios calculated by equation (III), see section 4.2, are also shown, for which a p$K_a$ = 3.16 was used (weighted average of all shown data at pH >2.4). The y-error bars represent the precision errors (2 $\sigma$), which are only visible for the $NO_x$/HONO ratio, but smaller than the size of the symbols for HONO and $NO_y$. The x-error bars represent the accuracy of the pH measurements.**


**Figure 3: NO and HONO concentrations of the HONO source (logarithmic scale) for variable nitrite concentrations (numbers in the figure in mg L$^{-1}$). The HONO signal of the chemiluminescence instrument shows significant tailing for zero measurements, which follow high HONO concentrations (see last zero). Experimental conditions: T = 15.9 °C, $\phi_{g,dry}^{\emptyset}$.= 2104 cm$^3$ min$^{-1}$, 20 rpm, pH = 2.48.**

**Figure 4: Nitrite-concentration dependence of the $NO_y$, HONO and $NO_x$ formation of the HONO source for two independent experiments. A) All data including the data shown in Figure 3; B) same data for nitrite concentrations ≤0.2 mg L$^{-1}$ (T = 15.9 °C, $\phi_{g,dry}^{\emptyset}$.= 2104 cm$^3$ min$^{-1}$, 20 rpm, pH = 2.45 and 2.48). The error bars represent the 2 $\sigma$ precision errors, which are smaller than the size of the symbols in A).**

**Figure 5: Gas flow rate dependence of $NO_y$, HONO and the ratio $NO_x$/HONO of the HONO source ($NO_2^-$ = 0.4 mg L$^{-1}$, T = 15.9 °C, 20 rpm, pH = 2.47). The error bars represent the precision errors (2 $\sigma$), which are only visible for the $NO_x$/HONO ratio, but smaller than the size of the symbols for HONO and $NO_y$.**

**Figure 6: Nitrite liquid flow rate dependence (10, 20, 30, 40 rpm of the peristaltic pump) of $NO_y$, HONO and the ratio $NO_x$/HONO**
**of the HONO source ($NO_2^-$ = 0.4 mg L$^{-1}$, T = 15.9 °C, $\phi_{g,dry}^{\emptyset}$.= 1570 cm$^3$ min$^{-1}$, pH = 2.47). The error bars represent the precision errors (2 $\sigma$), which are only visible for the $NO_x$/HONO ratio, but smaller than the size of the symbols for HONO and $NO_y$.**

**Figure 7: Temperature dependence of $NO_y$, HONO and the ratio $NO_x$/HONO of the HONO source ($NO_2^-$ = 0.8 mg L$^{-1}$, $\phi_{g,dry}^{\emptyset}$.= 1570 cm$^3$ min$^{-1}$, 20 rpm, pH = 2.46). The y-error bars represent the precision errors (2 $\sigma$), which are only visible for the $NO_x$/HONO ratio,**
**but smaller than the size of the symbols for HONO and $NO_y$. Accuracy errors of the temperature of ±0.3°C are also shown.**

**Figure 8: Stability tests of the HONO source at different HONO mixing ratios. On the left axis the chemiluminescence data of a high concentration experiment ($NO_2^-$ = 0.8 mg L$^{-1}$, T = 15.9 °C, $\phi_{g,dry}^{\emptyset}$.= 1570 cm$^3$ min$^{-1}$, 10 rpm, pH = 2.47) and on the right axis the LOPAP data at lower concentrations ($NO_2^-$ = 0.01 (0.001/0.004/0.002) mg L$^{-1}$, T = 15.9 °C, $\phi_{g,dry}^{\emptyset}$.= 2104 cm$^3$ min$^{-1}$, 20 rpm, pH =**
**2.54) are shown. For zero measurements the source was operated under water.**

**Figures**

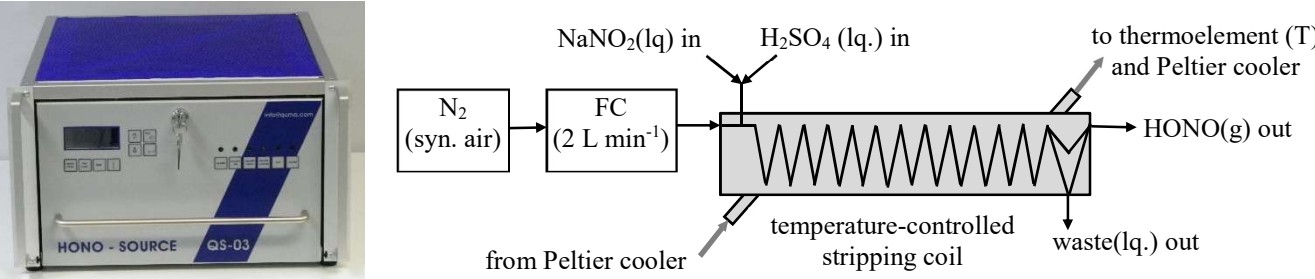

**Figure 1**

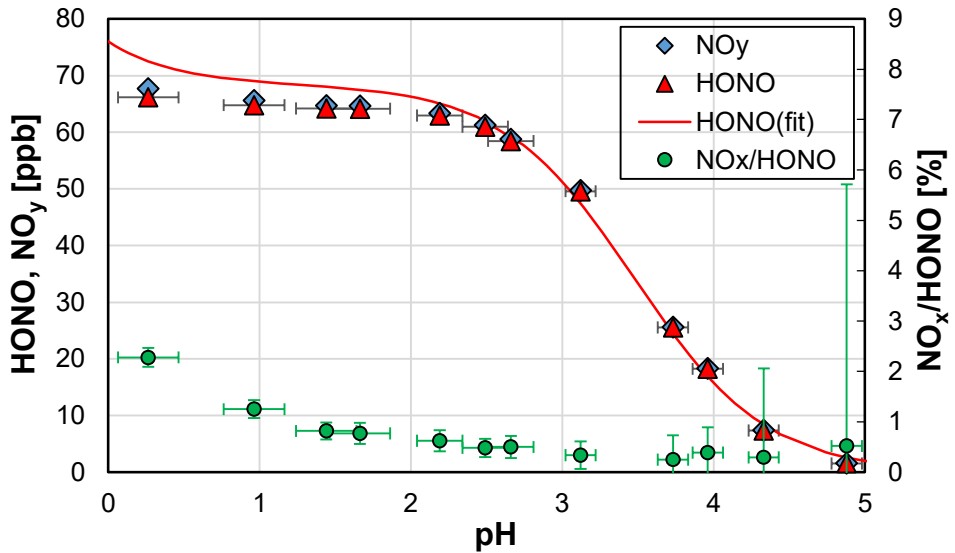

**Figure 2**


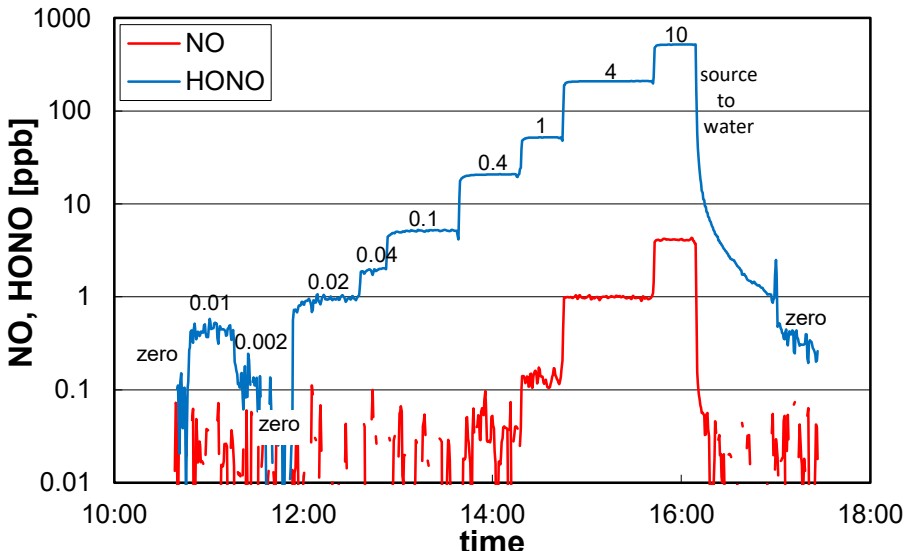

**Figure 3**

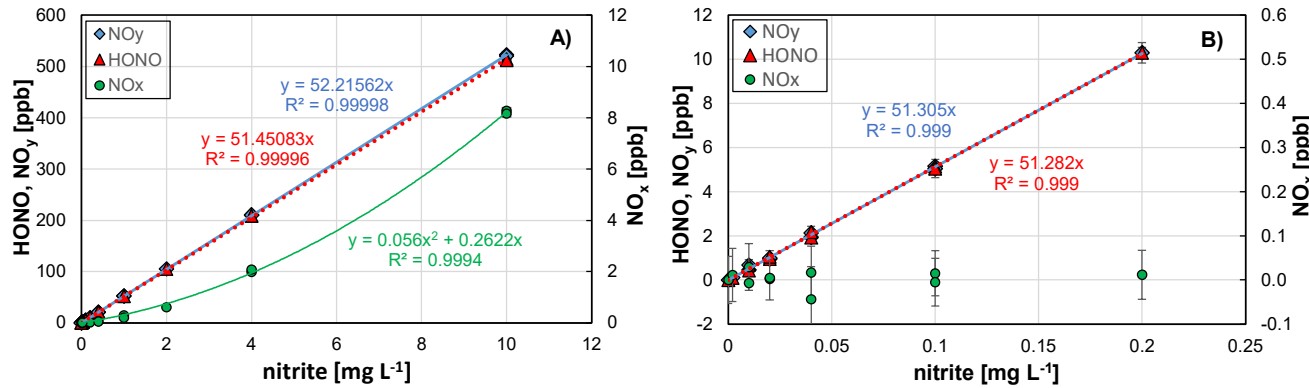


**Figure 4**

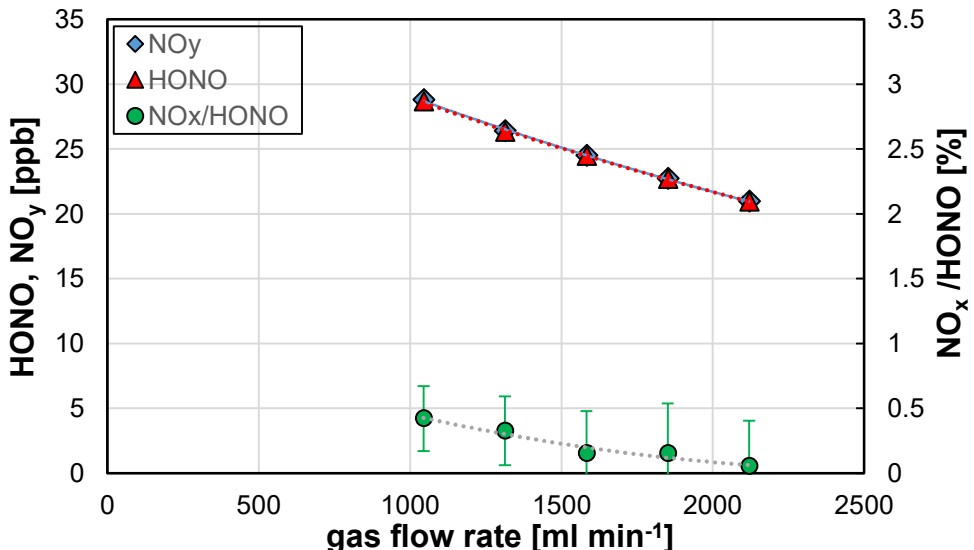

**Figure 5**


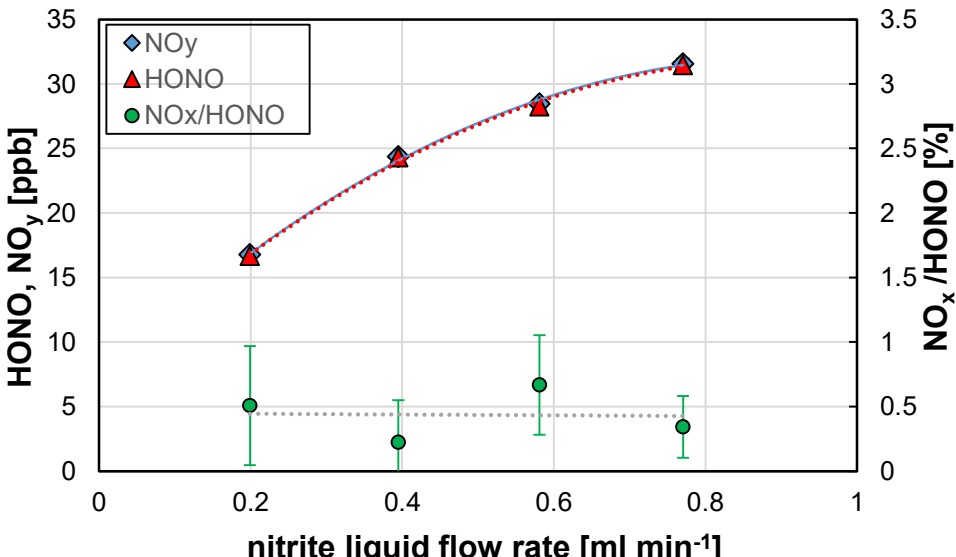

**Figure 6**

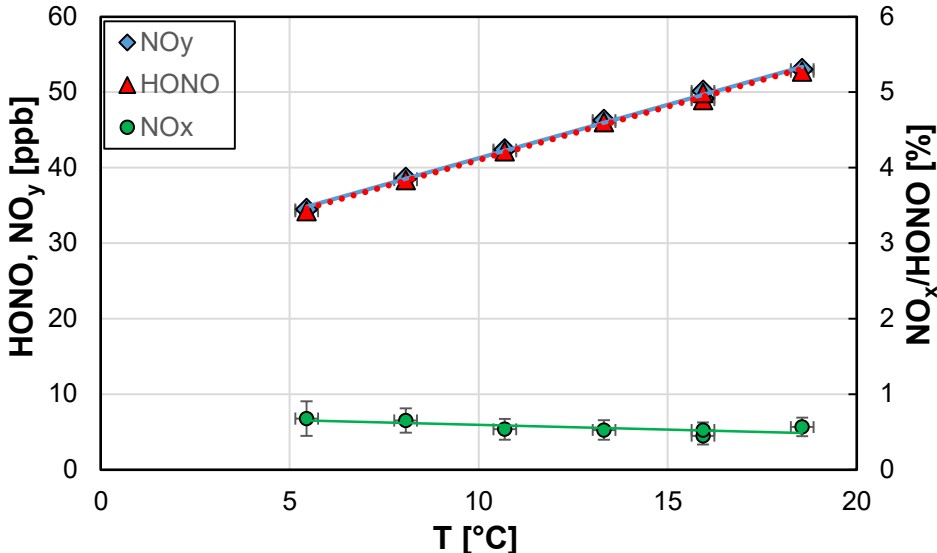

**Figure 7**

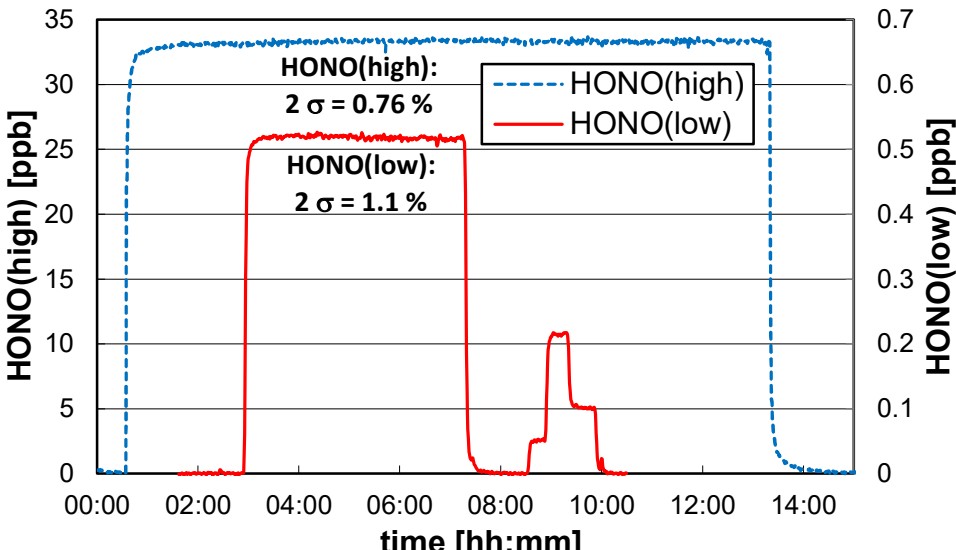

**Figure 8**