# Peer review of "A Source for the Continuous Generation of Pure and Quantifiable HONO Mixtures"

_Atmospheric Measurement Techniques, 2021_

## Author Response (AR1)

**Reply to Anonymous Referee #1**

*The manuscript "A Source for the Continuous Generation of Pure and Quantifiable HONO Mixtures" by Villena et al. reports the setup and the characterizations of a continuous source generating pure HONO mixtures. The manuscript is well organized and provides valuable information which is quite helpful for performing accurate HONO measurement. I recommend the publication. And only some minor comments follows.*

We would like to thank referee #1 for her/his interest and helpful comments, which will improve the quality of the paper.

*1. Line 81, Page 3: Would the 99.999% pure liquid nitrogen contain some NOx and contribute to the NOx formation of the HONO source?*

Answer: Since the used gas phase nitrogen was produced by evaporation of liquid nitrogen "5.0" (99.999 % purity) at the liquid nitrogen tank of the chemistry department and transferred to the laboratory by the central nitrogen feed line, the purity of the gas phase should be significantly better than 99.999 %, since most impurities (especially any $NO_2$ as HONO precursor) should remain in the liquid phase at the low temperatures (77 K) during the evaporation. In addition, from our experience with the purity of compressed nitrogen gas from the same manufacturer, we did not yet observe any measurable $NO_x$ impurities, excluding significant additional HONO formation.

*2. Line 87, Page 3: The generated gaseous HONO is guided to measurement instrument through PFA line. What about the wall loss of HONO on the line? It would be helpful if the authors provide a recommendation on the maximum length of the guiding line.*

Answer: Especially at low HONO levels we observed a significant tailing of the signals (see Figure 3), which we attributed to adsorption of HONO on the surfaces behind the HONO source. However, from the experiment shown in Figure 3 we conclude that most adsorption took place on the stainless-steel and filter surfaces of the inlet of the chemiluminescence instrument, but not on the PFA transfer lines used. At the end of the experiment shown in Figure 3, the source was first switched to water (at 16:09), which should make the HONO source to a perfect "zero-gas generator" after a short time. However, after a first fast decrease of the HONO mixing ratio (logarithmic scale…), there was a significant tailing of the signal at lower HONO levels. We think that this tailing does not results from the HONO source or the PFA surfaces behind the source, as the slope of the decreasing HONO levels did not significantly changed, when the HONO source was physically disconnected and the chemiluminescence instrument was operated by pure nitrogen (at 16:59). Possibly, HONO adsorbed on the inlet particle filter or on the stainless-steel inlet surfaces of the instrument is still desorbing to the gas phase even after longer time. This was also the reason, why the time response of the source slightly increased with decreasing HONO levels. In contrast, we do not think that HONO losses by adsorption on PFA surfaces is a significant problem. In former unpublished experiments with the HONO source at a smog chamber the exchange of a 3 m long PFA-line (4 mm i.d., 2 L/min) by a 20 m long one did not change the measured HONO levels, even at low ppb levels.

With respect to a similar comment by referee #2 we have added the following information to section 3.2, where Figure 3 is explained: "The increasing time response at low HONO levels is explained by adsorption/desorption of HONO on the surfaces behind the HONO source, which gets less important with increasing HONO levels, leading to faster saturation of the surfaces. From the experiment shown in Figure 3, we conclude that most of this adsorption/desorption took place on the surfaces of the chemiluminescence instrument used (inlet particle filter, stainless-steel lines) and not on the PFA transfer lines. At 16:09 the HONO source was switched from reagents to pure water, for which the HONO emissions should quickly decrease to zero. However, after a first fast decrease of the HONO concentration there was a significant tailing of the signal. Here the slope of the decreasing signal did not change when the HONO source was replaced by pure nitrogen at 16:59. This can only be explained when the tailing is caused by desorption of HONO from the surfaces of the chemiluminescence instrument, as all other PFA surfaces were removed. This conclusion is also in agreement with our experience with pure HONO mixtures, for which adsorption losses in PFA transfer lines of up to 20 m length were found insignificant."

*3. Line 179 - 181, Page 6: Please add in the figure caption what the error bars represent for. The same for other figure captions in the manuscript.*

Answer: The y-error bars in Figure 2 represent the precision errors (2 σ), which are only visible for the $NO_x$/HONO data, but smaller than the size of the symbols for HONO and $NO_y$. The x-error bars represent the accuracy of the pH, which was estimated to ±0.2 for pH<2, ±0.15 for pH 2-3 and ±0.1 for pH >3, caused by the problems with the pH measurements at higher acidity, see main text. In the revised manuscript all error bars will be explained in the figure captions.

*4. Line 184 - 188, Page 6: Please note that the time response indicated by Figure 3 should also contain that of the chemiluminescence instrument.*

Answer: Yes, we agree, most probably the decreasing time response of the source at lower HONO levels is caused by adsorption/desorption on the inlet surfaces of the chemiluminescence instrument, which will be explained in detail in the revised manuscript, see our response to point 2.

*5. Figure 3, Page 7: Concerning the stronger fluctuation of NOy signal observed at lower nitrite concentrations, does it caused by the measurement sensitivity not good enough or by the HONO source not stable under such condition?*

Answer: The noise of the HONO signal at low levels will be caused by the noise of the chemiluminescence instrument. Have in mind that the HONO signal is calculated from the difference of the signals of $NO_y$ and two times of NO, leading to corresponding higher noise of the calculated HONO data compared to the measured NO. In addition, the visually high noise at low concentrations is also caused by the logarithmic scale of the Figure. In contrast, there is no reason, why the emission of the HONO source should get more variable at lower HONO concentrations, since nothing (gas and liquid flow rates, mixing of the reagents, etc.) except the nitrite concentration was changed in the experiment.

*6. Section 3.6, Page 9: The stability of the HONO source is given by a 2σ relative error. Does it mean that the stability depends on the generated HONO concentration? However, when look at Figure 3, the fluctuation seems much smaller for higher HONO concentrations. Moreover, since ambient HONO concentration are mostly less than 10 ppb, information on the stability under such conditions would be even more valuable.*

Answer: As explained in the manuscript, the precision of the data shown in Figure 8 may result from the precision of the chemiluminescence instrument and the HONO source and therefore is specified as an upper limit. Since the relative error of the chemiluminescence instrument certainly decreases with increasing mixing ratios, indeed the HONO data at higher levels show an even better precision, e.g. of only 0.4 % for the 4 mg/l data in Figure 3. Since we did not systematically study the contribution of the $NO_x$-monitor to the precision errors of the measurement data, we simply quantified the upper limit error of both, the source and the $NO_x$-monitor from the experiment shown in Figure 8. However, since the variability of the HONO source should not depend on the HONO level (see answer to point 5.), we are quite confident that the given upper limit precision error is representative for all experimental conditions. Unfortunately, for the present study we made no long-time experiment at lower HONO levels, for which however the precision error would get higher caused by the lower precision of the $NO_x$ monitor at lower HONO levels.

*7. I fully agree with the authors' statement on the advantage of the reported HONO source over the traditional calibration method for instrument based on wet chemical techniques. Since the authors have LOPAP instrument in their lab, it would be helpful if the authors can provide comparison of ambient measurement results calibrated by gaseous HONO and liquid $NO_2^-$ standards.*

Answer: Here, the referee misunderstood our conclusion. First the source is ideal for calibration of HONO instruments, for which no simple calibration is possible, e.g. mass spectrometers. Second, for wet chemical techniques the source can be in addition used to characterise instruments, which are under development, e.g. to quantify their sampling efficiencies. However, after an instrument is well characterized, e.g. like the LOPAP instrument, the best calibration available should be used. For the LOPAP technique the accuracy error, when the instrument is classically calibrated by the liquid nitrite standard, can be optimized down to ±3-4 %, which is still lower, than the accuracy error of the HONO source (<10 %, ideally, ca. 6 %, see section 4.2.). For example, when we compared our LOPAP instrument for pure HONO/$NO_x$ mixtures in the EUPHORE smog chamber in two separate campaigns, deviations of <3 % were observed (see Figure 1 in Kleffmann et al., 2006). This high accuracy can not be obtained by calibration with the HONO source. Thus, we will not regularly calibrate our LOPAP instrument by the HONO source, but by the classical liquid calibration. However, this conclusion may be different for other wet-chemical instruments.

**Reply to Anonymous Referee #2**

*A continuous source for the generation of pure HONO mixtures from the sub-ppb range up to 500 ppb is developed and*
100 *characterized. nitrite is almost completely converted into HONO due to the acidic conditions of the aqueous phase (pH ≈ 2.5).*
*The source shows a fast time response of ~2 min (0-90 %) at higher concentrations and an excellent long-time stability. A*
*general equation based on Henry's law is developed, whereby the HONO concentration of the source can be calculated using*
*measured experimental parameters, i.e. nitrite concentration, liquid flow rates, gas flow rate, pH of the solution and*
*temperature of the stripping coil. For the calculation of the effective Henry's law constant, the acid dissociation equilibrium*
105 *of HONO/nitrite is used as a variable to adjust the theoretical HONO concentration to the measured values. A standard*
*deviation between all measured and theoretical HONO concentrations of only ±3.8 % is observed, for the first time, a stable*
*HONO source is developed. I believe this study is of great interest to readers. There are some comments that the authors*
*should consider, then the manuscript can be accepted to publish.*

We would like to thank referee #2 for her/his interest and helpful comments, which will improve the quality of the paper.

110

*1. Lines 161, " the theoretical and not the experimental pH values were used for pH <2 in Figure 2" Why use theoretical values instead of experimental pH values here?*

Answer: As explained in lines 157-161 of the manuscript, at pH<2 we were not able to measure the theoretical pH-values, expected assuming a reasonable quantitative dissociation of the strong sulfuric acid. Here, the measured pH was significantly
115 larger than the theoretical values. This deviation is a known artefact for electrochemical glass electrodes when using strong acids ("the acid error": Bates, 1973). In contrast, for pH>2 theoretical and experimental pH values agreed very well in the present study. Since quantitative dissociation of the strong sulfuric acid can be assumed in the pH range 0-2, we decided to use the theoretical values here. However, for the present HONO source, this artefact is no issue, since a fixed pH of ca. 2.5 is recommended, for which measured and theoretical pH agreed well (e.g. for the measured pH of 2.49 in Figure 2 the theoretical
120 value was 2.44).

*2. Lines 200, in figure 3, HONO should be added an ordinate.*

Answer: Thanks for pointing to this error. In a first version of this figure, we originally showed the measured $NO_y$ signal and not the calculated HONO ($NO_y$ - 2xNO). The ordinate title will be changed to "NO, HONO [ppb]".

125

*3. Lines 271, the English usage in the statement of " A HONO source was developed and characterized, where HONO is produced by the reaction of diluted nitrite and $H_2SO_4$ solutions in a temperature-controlled stripping coil " is not understandable and the sentence should be rephrased.*

Answer: We rephrased the sentence as follows: "In the present study a new HONO source was developed and characterized.
130 In contrast to most recent studies (Ren et al., 2010; Reed et al., 2016; Gingerysty and Osthoff, 2020; Lao et al., 2020), HONO is produced by the reaction of nitrite and $H_2SO_4$ in the liquid phase. In a stripping coil reactor HONO partitions to the gas phase according to its known moderate solubility in acidic solutions."

*4. Lines 280, why the time response depending on the HONO concentration levels?*

135 Answer: Indeed, the time response of the source slightly increased with decreasing HONO levels. The effect was however only observed at concentrations in the very low ppb range (see Figure 3). A possible explanation for this observation is the adsorption of HONO on humid surfaces behind the HONO source (glass surfaces of the exit of the stripping coil, PFA lines, PFA-T, inlet of the chemiluminescence instrument), which leads to some delayed response of the $NO_y$ signal. It is well known, that adsorption of gases plays a larger role at lower concentrations, as the time needed to saturate the surfaces increases at low
140 concentrations. E.g. at the end of the experiment shown in Figure 3, the source was first switched to water (at 16:09), which should make the HONO source to a perfect "zero-gas generator" after a short time. However, after a first fast decrease of the HONO mixing ratio, there was a significant tailing of the signal at lower HONO levels. We think that this tailing does not results from the HONO source or the PFA surfaces behind the source, as the slope of the decreasing HONO levels did not significantly change, when the HONO source was physically disconnected and the chemiluminescence instrument was
145 operated by pure nitrogen (at 16:59). Possibly, HONO adsorbed on the inlet particle filter or on the stainless-steel inlet surfaces of the instrument is still desorbing to the gas phase even after longer time. Thus, most probably this changed time response is an adsorption problem of the chemiluminescence instrument and not a problem of the HONO source. As reasons for this

observation are however not fully clear, we did not discuss this issue in the manuscript. However, it should be highlighted that even a time response of 7 min at a low HONO mixing ratio of 1 ppb is superior compared to any HONO source yet developed and we do not consider this issue too important.

With respect to a similar comment by referee #1, we have added the following information to section 3.2, where Figure 3 is explained: "The increasing time response at low HONO levels is explained by adsorption/desorption of HONO on the surfaces behind the HONO source, which gets less important with increasing HONO levels, leading to faster saturation of the surfaces. From the experiment shown in Figure 3, we conclude that most of this adsorption/desorption took place on the surfaces of the chemiluminescence instrument used (inlet particle filter, stainless-steel lines) and not on the PFA transfer lines. At 16:09 the HONO source was switched from reagents to pure water, for which the HONO emissions should quickly decrease to zero. However, after a first fast decrease of the HONO concentration there was a significant tailing of the signal. Here the slope of the decreasing signal did not change when the HONO source was replaced by pure nitrogen at 16:59. This can only be explained when the tailing is caused by desorption of HONO from the surfaces of the chemiluminescence instrument, as all other PFA surfaces were removed. This conclusion is also in agreement with our experience with pure HONO mixtures, for which adsorption losses in PFA transfer lines of up to 20 m length were found insignificant."

*5. In the part of 3.2, is it calibrated with Nitrogen ($N_2$) as the background? What is the gas flow in this part?*

Answer: As described in line 81 of the manuscript, the source was operated with pure nitrogen from our in-house nitrogen line for all experiments shown, but can be also operated with synthetic air. We used nitrogen here, as the nitrogen is expected to have smaller impurities compared to synthetic air. Although both have an original purity of better than 99.999 % (purity: "5.0"), the nitrogen is produced by evaporation from the liquid nitrogen tank of the chemistry department. It can be expected that most impurities (e.g. any $NO_x$) will stay in the liquid nitrogen at the low temperatures (77 K). In addition, the NO calibration gas used is also provided in pure nitrogen. Thus, the $NO_x$-instrument is exactly calibrated with the same buffer gas as used in the experiments, not affecting its sensitivity by any different quenching of the $NO_2^*$ formed in the chemiluminescence cell. In the experiments described in section 3.2, a nominal gas flow rate of 2 l/min was used for the flow controller of the HONO source, leading to a calibrated standard flow rate (298.15 K, 1 atm) of 2104 ml/min, see figure captions 3 and 4.

*6. In the part of 3.6, the source was operated at a low liquid pump speed of 10 rpm to get 2s noise is 0.76 %. Can you get the same value at the liquid pump speed of 20 rpm? Or the same value in the next experiment at 10 rpm?*

In this study, the long-time precision was only tested in the experiment shown in Figure 8, for which a liquid pump speed of 10 rpm was used. However, years ago we made similar tests with 20 rpm using a HONO-LOPAP instrument to quantify the output of the source and found a similar precision of 1.0 % (the slightly higher value is most probably caused by the lower precision of the HONO LOPAP compared to the chemiluminescence instrument used in the present study). Furthermore, when looking to the short-time precision during the liquid flow rate dependence (see section 3.4, with each step only ca. 30-45 min) the precision was found to be independent of the liquid flow rate. Also, for the longest HONO step shown in Figure 3 at a much higher HONO concentration compared to Figure 8 (with a corresponding better relative precision of the $NO_x$ instrument), an even better 2 σ precision of only 0.4 % was observed at 20 rpm, (although again at a much shorter duration of only 50 min). And finally, with respect to the second question, the same high precision is also obtained when the source was operated on two different days by two different operators, for which a mean deviation between the two sets of experiments of only 0.67 % was observed for HONO concentrations >20 ppb (see section 3.2 and Figure 4). Thus, the given upper limit precision error, which will also result from the precision of the chemiluminescence instrument should well describe the general stability of the HONO source.

**Reply to Anonymous Referee #3**

*Overall Comments*

*This manuscript provides the operational details of a commercial nitrous acid (HONO) calibration source that has previously been undescribed in the literature to allow replication and validation. As such, it has not been widely used for the calibration of in-situ atmospheric instrumentation. A nice instrumental intercomparison to show the utility of the source is made with the pairing of a chemiluminescent NOx monitor and a LOPAP, particularly because they are calibrated orthogonally. A stable and pure HONO source is demonstrated in the 10s of ppbv mixing ratio range along with tunability, but operational validation at mixing ratios relevant to the atmosphere are either neglected for discussion entirely or glossed over. It is confusing why the Authors have enumerated so many points on the excellent performance of the calibration source, drawing off their extensive measurement experience, to emphasize that those values apply at concentrations that are an order of magnitude (or more) higher than would be delivered for the operating range of an instrument (assuming no more than ten-fold dilution). Subject to inclusion of that information, since it has been collected, the manuscript is fit for publication in AMT once minor and technical revisions have been made.*

We would like to thank Referee #3 for her/his interest and the very detailed comments, which are addressed below point by point. The comments helped us to further strengthen our conclusions, by extending the measurement results to very low HONO levels.

In the overall comments, the referee criticized that the concentrations used in our study were an order of magnitude higher compared to the atmosphere. First, this is only partially true, since atmospheric concentrations of HONO reached up to 18 ppb in former studies (e.g. Los Angeles, Santiago de Chile, Milan), close to the 20-30 ppb used in most of our shown dependencies. Second, in the present study, also lower concentrations were used, for which for example in Fig. 4B the high linearity of the source was confirmed in the atmospheric concentration range 0.1-10 ppb. More importantly, for precise characterization of an instrument at least 10-20 times higher concentrations than the detection limit should be used (leading to a 5-10 % precision error, which may be just acceptable…). For the same reason, higher concentrations were used in the present study, caused by the sensitivity of the used chemiluminescence instrument. When working e.g. at 1 ppb, the precision error of the chemiluminescence instrument (and not of the HONO source, see below) would be too high (see the noise of the data in Fig. 3 at these low HONO levels). E.g. if working at only 10 times the detection limit of the chemiluminescence instrument, the purity of the source could not be better quantified than ±10%, not sufficient for our study, where a purity of 99.8±0.2% was experimentally obtained at 20 ppb HONO (see details below). Also from our experience with characterization of different HONO instruments, typically concentrations in the higher range were requested in the past, e.g. by the colleagues during the FIONA intercomparison campaign in the EUPHORE chamber (Rodenas et al., 2013). Here most optical instruments were not sensitive enough to work precisely at the low ppb range. In addition, when we use the source in our lab, e.g. to measure the HONO sampling efficiency of the stripping coil of a LOPAP instrument (typically >99%), HONO concentrations of tens of ppb are required to precisely quantify the <1 % loss from the coil. Furthermore, following general analytical rules when calibrating any instrument by a typical two-point calibration (with prior validated linear response), this should be done at the upper limit of the measurement range. For HONO this should ideally be around 20 ppb to cover all atmospheric concentrations (see above). Thus, the HONO source should be also operated at these higher levels. And finally, even higher than atmospheric HONO concentrations are often required in the laboratory, when for example, the UV absorption spectra of HONO is measured (see e.g. Stutz et al., 2000). This was the reason, why we extended our concentration dependency up to 500 ppb HONO. The application of the source is not only limited to the calibration of HONO instruments at fixed low ppb levels, but can be used for many different purposes (see also e.g. lines 346-347), which the referee may have missed here.

Besides these general remarks to the concentration range used, the precision of the present HONO source is not expected to change with the HONO concentration by any scientific reason. The stability of the HONO source is only depending on the mixing of the reagents at the inlet of the stripping coil (see Fig. 1) and the variations of the gas and liquid flows. These parameters will not change with the nitrite concentration and thus the relative stability of the HONO concentration - limited

235 by Henry's law ($p_g \propto c_{lq}$, see equation (I)) – will be independent of the concentration. This is confirmed by the perfect linear response of the source in the very typical atmospheric concentration range of 0.1-10 ppb (see Fig. 4B, with the same slope for HONO compared to all data up to 500 ppb…). Thus, working at 20-30 ppb will not "*gloss over*" the quality of the source, but is caused by the sensitivity of the chemiluminescence instrument used and the upper limit precision errors, which we wanted to accept here (and we consider a precision error of ±10% as used in other studies as too low).

240 However, to confirm the referee, we have done a similar experiment as shown in Fig. 8 at much lower HONO mixing ratio (~0.5 ppb), using the more sensitive LOPAP technique and not the chemiluminescence instrument (see figure below). For this low concentration data again a high precision of 1.1 % was obtained. The slightly lower precision compared to the high concentration data shown before (0.76%) is caused by the lower precision of the LOPAP. In addition, for a concentration dependency in the low concentration range of 0.05 – 0.5 ppb (see also Figure below) we obtained again an excellent correlation

245 of the mixing ratio with the nitrite concentration ($R^2 = 0.9997$), in agreement with the results shown in Figure 4. The high precision of the source and its absolute accuracy also at lower HONO levels is confirmed by the low average deviation between experimental and theoretical HONO mixing ratios (see equation (III)) of only 1.1 %. For this comparison even the pKa was not adjusted (see lines 146-147), but the average value determined in the present study was used (see line 149-150). Thus, the high precision and accuracy of the source was confirmed down to 50 ppt HONO. We will add this information and a new

250 Figure 8 (see below) to the revised manuscript.

[Figure]

**Figure 8: Stability tests of the HONO source at different HONO mixing ratios. On the left axis the high concentration chemiluminescence data (NO$_2^-$ = 0.8 mg L$^{-1}$, T = 15.9 °C, $\phi_{g,dry}^{\emptyset}$ = 1570 cm$^3$ min$^{-1}$, 10 rpm, pH = 2.47) and on the right axis the low concentration LOPAP data (NO$_2^-$ = 0.01 (0.001/0.004/0.002) mg L$^{-1}$, T = 15.9 °C, $\phi_{g,dry}^{\emptyset}$ = 2104 cm$^3$ min$^{-1}$,**

255 **20 rpm, pH = 2.54) are shown. For zero measurements the source was operated by water.**

The referee also mentioned in the overall comments that the HONO source was yet "undescribed", which is not correct. The principle of the source is based on the former bubbler set-up of Taira and Kanda (1990), see line 65, which we only modified by using a stripping coil. This modified source was already described in our former paper Kleffmann et al. (2004), see lines 70-72. However, since the source was not explained in detail with all the different dependencies in this former study and since

the source was indeed not recognized by the community in the past, we decided to publish details of the source in the present study, including its theoretical quantification. Our work was mainly motivated by the different recent studies on new HONO sources based on the Febo et al. principle, however with lower purity and stability compared to the original set-up. Since we also used the original Febo et al. source in the past, we know that the present stripping coil source has many major advantages, which we wanted to present to the community (see general discussion in section 4.1).

**Minor Revisions**

1. *Alphabetized lists are present throughout the manuscript and significantly detract from the quality of the points being made. In many places these are formatted badly and make the logic challenging to follow. In most cases, these can simply be replaced with a structured paragraph to address each point and some minor reorganization.*

We will reformat the listing as recommended, see below for details.

2. *The Authors are mixing metrics in their comparisons with other instruments in several instances that give misleading impressions on their performance. In most cases these can be corrected by clearly separating the terms being discussed (see technical corrections below).*

Find below the answer to the technical corrections.

3. *Present the performance metrics for the calibration source at output mixing ratios of 5 ppbv and below. This is the range that will be required to calibrate instruments for ambient measurements, as it will set instrumental accuracy and precision. It is critical to present the stability of the instrument that is applicable. Of course the metrics look great over 10 ppbv, just like every other high-output HONO source, but that applies to very few real situations (e.g. wildfire plumes or tunnels).*

As explained above, we will also show an experiment at lower HONO level to confirm the referee that the precision of the source is not a function of the HONO concentration (why should it?). In addition, 5 ppb will not cover the atmospheric concentration range of HONO, so we disagree to this general statement. You find 18 ppb HONO not only in tunnels or in wildfire plumes, but also e.g. in the urban background in Santiago de Chile (Elshorbany et al., 2010). In addition, the source can be used for very different purposes and is not limited to only calibrate a HONO instrument at fixed low HONO concentrations (for more details, see our answer to the overall comments above).

**Technical Corrections**

*Page 1, Lines 24-25: Source stability has been solved for a long time (since Febo, ~2%), so this point that it is the first is not accurate. The statement on providing the first absolute calibration source is accurate and the most noteworthy contribution of this work. Revise the statement to prevent it from being misleading.*

The statement "*for the first time, a stable HONO source is developed, which can be used for the absolute calibration of HONO instruments*" is correct. Besides the fact that this is indeed the first stand-alone calibration source (you do not need another HONO instrument to quantify the output, as with other sources), our precision is also superior. The precision is better than 2% (and also at sub-ppb levels, see above), which was already visible from Figure 4 B. Our precision is even much better than the lower precisions obtained for other recent HONO sources. For example, in Fig. 2/Tab. 2 of Lao et al. (2020), a precision of

only 15% was obtained at a HONO level of 2.8 ppb. Thus, the very high precision down to 50 ppt HONO (see the new figure in the overall comments) is another feature of the HONO source, which we would like to highlight by this sentence.

***Page 3, Lines 68-70: This is another misleading sentence that needs rewriting to accurately represent the current state of knowledge. Varying the temperature and HCl concentrations are required only once to identify the required working range of the system and are not varied once that fact has been established. There are no reports of this system being modified in real-time to change HONO output of a calibration system, except for proof-of-concept to help end users know their options for obtaining a desired HONO output. Further, varying temperature is trivial and permeation tubes are prolific across many industries for gas calibrations. The Authors are being disingenuous by calling either of these complicated, but are correct that each could require some time to either acquire (permeation tubes require certification by manufacturers) or reach a programmed setpoint with a PID controller (temperature). It is fairly standard practice to expect such time requirements from a calibration system, especially given the time that one must commit to pipette solutions into cleaned labware to set up the calibration source, then prime the peristaltic pump (and so on) in the work presented here.***

This section describes the history of the development of *our* HONO source and the cited sentence is only related to the *specific* source by Febo et al. We used this source already more than two decades ago in our laboratory, when our Italian colleagues visited us for a common study (Becker et al., 1995) and we later copied it for other studies (Brust et al., 2000). Thus, we very well know, how this source is working and confirm that the mentioned sentence "*In contrast, for the source by Febo et al. (1995), variations of the temperature and the concentration of the liquid HCl in the permeation source are required, which is more complicated and time-consuming.*" is correct und will not be changed. We did not used recent modifications of the Febo source, see "*accurately represent the current state of knowledge*" (see e.g. Gingerysty and Osthoff, 2020; Lao et al., 2020). How could that play a role during the development of our source (done in 2003, see history…)?

And again, the referee may have misunderstood the potential applications of the source, which is not only aimed to calibrate HONO instruments at a fixed HONO concentration ("*are not varied once that fact has been established*"), but also to use it for different applications in the lab (see above). Often the fast, predictable and precise change of the HONO concentrations by minimal modification (the Teflon nitrite feed line is dipped from one nitrite solution into another…) is highly welcome. For example, multipoint calibrations were necessary during the calibration of a CIMS instrument, which showed a highly non-linear HONO response (Jurkat et al., 2011). Here a source like that mentioned by the referee would not have helped. The fact that we could precisely change the concentration in a short time simply by changing the nitrite concentration (see Fig. 3 and Fig. 4) was very much appreciated by our colleagues from DLR. We could not dilute a constant HONO source by synthetic air and MFCs in these experiments (see recommendation below by the referee) caused by the changing humidity. The response of a CIMS instrument can also be humidity dependent, caused by different cluster formation (for further details, see below). And even if humidified air is used for dilution with MFCs, the maintenances of constant humidity is still a difficult task. In our source fast and predictable concentration changes can be obtained at absolutely constant humidity. So here the view of the referee to our study is too limited.

We have added in section 4.1 the following information: "*Dilution of the source by synthetic air and using flow controllers is not recommended, caused by the decreasing precision of the source and the resulting variable humidity. The latter can be a problem, when a humidity dependent HONO instrument is characterized, e.g. when the CIMS technique is used (Jurkat et al., 2011). In contrast, for the present source, variable HONO concentrations are obtained for constant humidity.*"

And finally concerning the requested time for these modifications ("*the time that one must commit to pipette solutions into cleaned labware to set up the calibration source, then prime the peristaltic pump (and so on)*"): the nitrite solutions can be made before or during experiments and thus the time for preparation, does not limit the time to change the HONO concentration in an experiment, which is discussed here. In contrast, in the Febo et al. source the time for exchanging (not for preparation…) the HCl solution (ca. 1 l in a temperature-controlled bottle…) or the time to change the temperature of the bottle followed by

the stabilisation of the HONO concentration takes much longer (at least 1-2 h). In our source the concentration change is done in 10 min.

***Page 3, Lines 85-87: This sentence is hard to follow. There is fragmentation and mixing of ideas. Revise into two sentences that are complete.***

340 We have split the sentence: "*The maximum temperature of the stripping coil is limited to a few °C below room temperature. If higher temperatures are used, water will condensate in the PFA (perfluoroalkoxy alkanes) lines (4 mm i.d.).*"

***Page 3, Lines 94-95: The elasticity of peristaltic tubing degrades over time and can lead to poor flow control or a total loss of flow. Can the Authors please add some instruction on this for the presented system to the discussion where they are commenting on the volume requirements for the calibration source, so readers have an idea of the necessary maintenance?***
345 ***Can the Authors also provide some sort of objective metric to identify that the peristaltic tubing may be compromised in function?***

The used three-stopper long-life tubes (see line 96) have a lifetime of ca. 1000 h on each of the two positions and the whole tubes are exchanged before that lifetime (ca. 2000 h) is reached, which is a standard procedure when using a peristaltic pump. Since this is done in a few minutes and since the costs for this maintenance are minimal, this is not an important issue. However,
350 the referee is correct, flow rates of peristaltic pumps typically slightly decrease with time (ca. 20 % over the whole lifetime) and thus the liquid flow rates should ideally be measured on the same day, when accurate quantification of the HONO concentration is necessary. We will add this information in the experimental part of the revised manuscript.
In contrast to this long-time drift of the flow rate, there is also an initial change of the flow-rate, when the source is started. This is caused by the warming up of the Ismaprene tubes during ca. the first hour. For this reason, we recommend that the
355 source is started under water. During operation with water, it can be already used to zero a HONO instrument, since a variable water flow rate will not affect the zero output and the humidity of the source. After one hour, water is exchanged by the reagents and after a short time the HONO source can be used for any application. This recommendation is already explained in lines 287-289.

***Page 4, Line 128: This list is not necessary. Write with full sentences and paragraphs.***

360 As suggested, we will change that section using full sentences to list the different points.

***Page 4, Line 131: Units should be given in parentheses. Also, why is the molar concentration of HONO not presented as [HONO] in Equation (I)?***

The units are given in parentheses [unit] throughout the whole manuscript, so we do not understand that issue? In addition, we prefer using the more precise liquid concentration term $c_{lq}$ than the one the referee recommends, see also equation (III)
365 for the nitrite concentration. To unify the terms we changed the term $[H^+]$ in equation (II) to $c_{H+}$.

***Page 5, Lines 157-159: There are too many ideas intermixed in this sentence. Please revise into two or three sentences for clarity.***
Sorry for this complicated section, which we modified to:
"*One potential problem could be the pH measurements, which showed excellent agreement between measured and theoretical*
370 *values only for pH >2. In contrast, at higher acidity measured pH-values were significantly higher than theoretically expected, which is a known problem when using pH-electrodes (Bates, 1973). Thus, the theoretical and not the experimental pH-values*

*were used for pH <2 in Figure 2. For calculation of the theoretical pH, the acid concentration was used and reasonable quantitative dissociation of the strong $H_2SO_4$ was assumed.*"

**Page 6, Lines 196-199: *What about below 5 ppbv output? It's been long demonstrated that it is easy to get reproducible and stable HONO outputs for sources with high concentrations. These are not found in the real atmosphere, so the instruments would not be calibrated in their working range. It is also not reasonable to perform subsequent dilutions greater than a factor of 10 with MFCs as most users will not commit to the very large gas requirements (or potential pressure issues).***

First, with our source no dilution with MFCs is necessary and also not recommended (see above: non-stable humidity). Caused by the perfect linearity of the source and the simple adjustment of the concentration by varying the nitrite concentration (see above), only one MFC with a very typical gas requirement of 0.5-2 l min$^{-1}$ is used (see experimental section and Fig. 1). In addition, dilution using at least two MFCs would lead to higher precision errors, which we want to avoid. Second, exactly for the reason mentioned by the referee, we have shown the slopes in Fig. 4 not only for the whole dataset (see Fig. 4A: 0.1-500 ppb), but also for the data obtained at the very reasonable atmospheric concentration range of 0.1-10 ppb (see Fig. 4B). Since the slopes were the same for HONO in both plots, the high precision of the source is not only obtained at high concentrations, but also at the requested atmospheric HONO level of <5 ppb. To further confirm the referee, we will add the modified Fig. 8 (see above), where the very high precision (ca. 1 %) was obtained also at rural HONO levels in the range 0.05 – 0.5 ppb.

**Page 7, Lines 210-211: *Clarify that the NOx monitor requires at least 1 L min-1 here. The way this is written makes it seem like the instrument can only handle flows of 1 L min-1, but a higher flow could be directed to it with an appropriate atmospheric vent or waste line used for the excess gas.***

This is explained in the experimental section, see lines 104 and 113-115 and at the beginning of section 3.3. An excess air flow of the source over the flow rate of the NO$_x$-instrument (1 L min$^{-1}$) is obviously necessary to ensure no dilution of the source by ambient air by the gas vent. However, the source can be also operated at flow rates down to 0.5 L min$^{-1}$ (see line 83). At lower gas flow rates the stripping coil does not work well. To clarify we have added to the end of the experimental section: "*Caused by this set-up, the lower flow rate of the HONO source applied in the present study was limited to the flow rate of the chemiluminescence instrument (1 L min$^{-1}$).*"

**Page 8, Line 233: *The point here is that higher flows generate higher concentrations of HONO, which can then decompose to NOx. Revise this sentence so it does not seem that a separate issue of NOx production that depends on flow exists (e.g. due to turbulent flow dynamics).***

Our HONO source has the same problem as any HONO source, leading to some heterogeneous decomposition of HONO to NO$_x$ (back reaction (1)), which shows a well-known quadratic concentration dependence (see Fig. 4). When decreasing the gas flow rate, the decomposition of HONO at longer contact time in the system leads to increasing heterogeneous NO$_x$ formation (see Fig. 5). In contrast, for any homogeneous decomposition in the liquid phase, the NO$_x$ content would not change with the gas flow rate (at constant liquid flow rate). In Fig. 6, to which the referee refers to, we show that the variation of the liquid flow rate has no impact on the NO$_x$ content. Considering that both, the surface area and the gas/surface contact time in the system do not change, the heterogeneous nature of the reaction is again confirmed. In contrast, for any potential homogeneous liquid phase decomposition of HONO, increasing liquid volume in the stripping coil (at higher liquid flow rates) would lead to increasing NO$_x$ content. Since this is not observed, a homogeneous decomposition can be excluded. This is an important observation, which we described in this section and which we would like to leave here. In order to describe more precisely our observation in the figure we will changed "*NO$_x$-formation*" to "*NO$_x$-content*" in line 232.

410    In contrast, the time response of the source (this is the main sentence in the cited line 233, but not mentioned by the referee?) is depending on the liquid exchange rate.

*Page 9, Lines 244-245: As a standalone statement about RH, I do not understand the value of having a water vapour saturated in the calibration gas flow. If anything, this is a problem. The Authors point out that problem as well, saying that condensation can occur if the ambient temperature is below that of the dew point compared to the stripping coil. I'd suggest*
415    *removing this sentence or making a clear point on why the water vapour in the calibration flow is useful.*

As confirmed in our study (see lines 321-324) the absolute humidity of the source is given by the temperature of the stripping coil, where the gas phase is saturated (near to 100 % r.h. at the temperature of the coil, see also Fig. 1). However, since the saturation water vapor pressure is temperature dependent according to the Clausius Clapeyron equation, the absolute humidity is decreasing with decreasing temperature of the stripping coil. This is explained in the sentence: "*Besides the HONO levels,*
420    *also the humidity of the gas phase can be varied by the temperature of the stripping coil.*" In the revised manuscript, we will modify to "*absolute humidity*" and will add "*(see Clausius Clapeyron)*". But since the stripping coil is always operated at lower temperatures than room temperature (see lines 85-87), the gas exiting the source will be never saturated (at room temperature of the transfer lines…), so this is not a problem. Since the simple variation of the humidity is another important advantage of our source (see also lines 300-301), we will not remove the sentence. The relative humidity of our source can be
425    adjusted to most relevant atmospheric conditions (35-80 % r.h. at 20°C room temperature), which is often necessary, e.g. when calibrating a HONO instrument with variable humidity response, like the CIMS system mentioned above. If lower humidity than the dew point of 5 °C (see line 84) is required, the HONO source must be diluted with dry synthetic air. This is easily possible for lab studies, but is not recommended for calibration purposes, because of the increasing precision errors (see above).

*Page 9, Lines 247-248: Does this imply that the backreaction is exothermic? Such that the elevated temperature is reducing*
430    *the decomposition pathway to NOx despite the higher HONO being produced? I'm not sure the current statement is accurate in terms of explanatory power despite the observed relationships being correctly stated.*

Possible explanations for the slightly decreasing $NO_x$ content of the source at higher temperature are either the thermochemistry of the back reaction (1), as the referee suggests, or more reasonably, the amount of water adsorbed on the surfaces behind the source. Here higher humidity will push equilibrium (1) to HONO leading to the observed slightly lower $NO_x$ content at higher
435    temperature (and humidity). However, since we did not study this decomposition reaction in detail, which is out of the scope of this instrument's characterization study, we did not give any explanation. In addition, we do not consider the $NO_x$ formation to be too important, since it is not significant under most conditions and were only statistically significant here caused by the relatively high HONO levels used. The $NO_x$-impurities were of the order of 0.5 % during this experiment, which can be further reduced when working at lower HONO levels (see Fig. 4) and when using a flow rate of 2 L min$^{-1}$ (see Fig. 5).

440    *Page 9, Line 259: What about the variance at lower mixing ratios? This evaluation is way above even the highest ambient mixing ratios observed in the real world, excepting extreme cases like wildfire plumes. The duration over which the variance was determined (as presented in Figure 8) also seems to be selected arbitrarily, rather than reflecting a typical span of time that one would conduct a calibration over (e.g. 1-2 hours). As a result, the additional data points from ~12 hours of observations make the variance seem much smaller than the more relevant timescale. It would be more useful to see the*
445    *application-relevant performance of this calibration source at 5 ppbv, 2 ppbv and ~100 pptv, as suggested is possible in Figure 4B.*

As already explained above in the answer to overall comments, the precision of the source will not depend on the concentration, caused by the use of the Henry's law principle. To confirm we will show a similar experiment at lower concentration in the

revised manuscript. Concerning the second point that the long time period used may improve the observed stability: for most data points, shown in the manuscript, the mentioned 1-2 hour periods (or even shorter periods, see e.g. Fig. 3) were used. Also, with these shorter periods we still obtained a very high precision, see e.g. the high linearity of the source shown in Fig. 4. In addition, for all variable experimental conditions, the ratio of predicted and observed concentrations varied by only 1.7/3.8 % (see lines 328-331) when using all data >5 ppb/all data including the less precise low concentration chemiluminescence data. The slightly lower precision when all data >5 ppb is considered (1.7%) compared to the precision of 0.76% shown in Fig. 8. is explained by the very variable conditions applied for the first. Thus, the specified precision clearly refers only to constant experimental conditions of the source (liquid flow rate, gas flow rate, temperature).

*Page 9, Lines 262-263: This continuous duration is shorter than reported for other sources. That should be stated clearly and the point that this system can be shut down, flushed with deionised water, and restarted easily emphasized. The major contribution of this work is quantitative HONO production by mixing the two reagent solutions together on-demand with very little stabilization time required. It would be nice to see a depiction of that 'start-up' from instrument measurements over which the HONO is produced rapidly and with high stability. Apologies if this is what is being shown in Figure 8, but the initial and final conditions are not clearly stated as having changed the nitrite solution for deionised water or articulating a valve to deliver clean air to the instrument instead of the HONO source flow.*

As the referee stated, our source is normally not used continuously for several days or weeks as it provides stable HONO mixtures already after short time periods. It can be switched on, when necessary, also minimizing air and liquid consumption (see point of the referee above to Lines 196-199) and used after ca. one hour of flow stabilisation. Even in the FIONA intercomparison campaign (Rodenas et al., 2013), where the source was in use all the two weeks, we switched it on every day again. Therefore, in Fig. 8 we showed only one night of use, which is sufficient for our typical applications.

In contrast, for other recent HONO sources longer stabilization times are necessary and thus, the source should also be tested for longer time periods. E.g. in Lao et al. (2020), weeks were already necessary for the stabilisation of only the HCl permeation source. In a field campaign, lasting only 2 weeks, this may be difficult and is only possible when the HCl permeation source in continuously working also during transport (operated by a battery?).

However, in contrast to the statement given by the referee, there is no reason against longer usage of our source, e.g. for two weeks, when larger reagent/waste containers are used (see line 263).

Finally, we accept the apologies, because that is exactly shown in Figure 8. The HONO source was started under water and switched to reagents at around 19:00. On the morning of the next day the reagents were again exchanged by water. This procedure was applied for almost all zero data shown in the manuscript. We will add these details to the caption of the revised Fig. 8.

*Page 10, Lines 275-276: This is true only over a few hours. The Authors state that pump flow rates drift over such periods, which means that they need to be recalibrated nearly daily. It is true that this task is simple, but one could argue that this is just as much of a malfunction as those observed in other sources. It would also be instructive to indicate how long the prepared nitrite solutions are stable for and under what conditions somewhere.*

No, that is true for ever, since we refer in the two lines to the possible unwanted formation of by-products (HCl, ClNO) when the chemistry of the Febo et al. source is used and not to any general malfunction, which any instrument may have (e.g. power supply failure in a field campaign etc.). By the chemical scheme used in our source (adopted from Taira and Kanda, 1990), these by-products are impossible by definition. Furthermore, the "pump drift" is only very low (20 % in 1000 h…) and not measurable "*in a few hours*" and is not of any importance, if the flow rate is determined on the same day when the HONO concentration has to be calculated (as recommended). This is not a malfunction? And if requested (but not necessary),

expensive liquid flow meters could be even used to automatically correct for these small liquid pump drifts. With respect to
490    the last comment, if nitrite concentrations of 0.1 mg/l or higher are used, these solutions are stable for weeks, when stored in
the dark. For the long-time stability of lower concentrations, we have no experience and recommend daily preparation. This
information will be added to the revised manuscript.

*Page 10, Line 281: The Authors are conflating changing the HONO output from their source with the ability to easily dilute*
*the HONO generated by others. Other sources can have their outputs rapidly modified on timescales of seconds using mass*
495    *flow controllers, so the comparison being made is not fair-minded. The initial range for other sources simply requires*
*proper setup. In fact, the Authors do not present good reasons to me why one would want to generate >50 ppbv of HONO*
*for calibration purposes? As such, why not identify the correct nitrite concentration to obtain an output that is easily diluted*
*into the ambient range of observed HONO mixing ratios? If a change in nitrite concentration in minutes then can give*
*access to even lower stable mixing ratios (<10 pptv), that would be very attractive.*

500    First, we have already mentioned different reasons (lower precision, variable humidity) why we do not want to dilute the
HONO source by MFCs and synthetic air (see above). Second, we also already explained, why concentrations of 500 ppb can
be necessary in the lab (in the work of Stutz et al., 2000 even ppms were applied…). Again, this source can be used for many
different purposes (see above) besides the mentioned simple calibration of a HONO instrument. With respect to the mixing
ratios <10 ppt: this should be theoretically possible, caused by the principle used (see the 50 ppt step shown above in the
505    revised manuscript), but would be challenging for several reasons. First, even sensitive HONO instruments with detection
limits around one ppt, should not be calibrated by a 10 ppt mixture (zero + span) with respect to general analytical rules (span
at the maximum measurement range). Second, systematic errors would be also much higher than specified, since several
dilutions steps of the nitrite solution would be necessary to generate such low HONO levels (i.e. only around 0.2 µg/l nitrite,
commercial stock solutions typically contain 1000 mg/l). Finally, at 10 ppt delayed response is expected in any system, caused
510    by increasing adsorption of HONO on surfaces, which is a general problem when working at ultralow trace gas levels.

*Page 10, Line 284: This work reports 2 hours for stability under the recommended operation conditions. Correct this*
*statement.*

This statement referred to Figure 3 of the cited study, in which a stabilization time of at least seven hours is shown (for the
blue data it is even longer). But since indeed the 2 hours are later specified, when the NaNO$_2$ devices were used already several
515    times before, we will correct this in the revised manuscript.

*Page 10, Lines 285-286: As stated above, permeation devices have been commercially produced for a very long time. They*
*are sold with stabilized outputs that are certified, which requires no stabilization time to use. There are reviews on this cited*
*in the work they are referencing. Here, the Authors are commenting on homemade permeation devices being stabilized on*
*much shorter timescales compared to certified commercial options (days instead of 6 weeks). One could easily produce*
520    *dozens of these at once and have stable HCl permeation devices for years, negating the statement made here. How long*
*does it take to order and prepare the reagents for this source? Why are those timelines not considered in this comparison?*
*Given the prevalence of permeation devices in use, the Authors are recommended to reduce their focus on this point, as*
*HCl emissions with 5% accuracy could easily be obtained from a commercial manufacturer in perpetuity, with only the*
*initial waiting period to consider.*

525    We have to apologize that we have no experience with commercial permeation devices and can thus only refer to those studies
in which permeation devices were used to produce HONO. And here we would like to refer to Figure S10 of the study of Lao
et al. (2020), where even after 15 days of use unwanted HONO peaks appeared in the source (see HONO steps in run 2 at 15

h and 21 h). For explanation of these peaks only HCl emission peaks are reasonable (see their Fig. 4b to confirm), since the NaNO$_2$ is present in excess and will be stable for that short period. Thus, as long as the statement by the referee (5% accuracy)
530 is not demonstrated using a commercial HCl permeation device, we can only refer to the numbers specified in the existing literature of HONO sources.

***Page 10, Lines 287-289: This is a separate discussion point, not a contrasting one. Move to a separate part of the discussion. As suggested above, it would be great to show this performance in action where water and nitrite are exchanged in real time to demonstrate the start up and shut down periods that the system can achieve.***

535 No, it is a contrasting point, since the time to start our source is much shorter than in any other known HONO source. And to the second point: As already explained above, this is exactly shown in Fig. 8.

***Page 10, Lines 292-294: For the Febo-style source, changing the temperature of the entire HCl solution is indeed time consuming, but it is easily done with a PID controller, so not particularly difficult. Also, if the relevant temperature to obtain HONO mixing ratios relevant to the operating range of an atmospheric instrument is identified, why would one be***
540 ***changing this regularly? They would simply be diluting the output with a mass flow controller and zero air. The Authors are putting a lot of emphasis on obtaining a wide dynamic range of HONO mixing ratios, particularly those above the observed atmospheric range. Why?***

See detailed explanations given above. Dilution by MFCs are not recommended (precision, humidity), multipoint calibration can be important and high concentrations are often necessary in laboratory studies (see e.g. Stutz et al., 2000 and Kleffmann
545 et al., 2004), etc.

***Page 11, Line 295: The NOx decomposition is still occurring, the mixing ratios are simply below the detection limit of the monitor. Revise for accuracy.***

No, that is not correct. First, we confirmed the quadratic NO$_x$ formation with increasing HONO levels expected from the heterogeneous decomposition kinetics of HONO on surfaces (see Figure 4). Thus, the impurities of NO$_x$ decrease with
550 decreasing HONO levels. Second, and more important, the measured purity of the source at 20 ppb, was 99.8 % in between the precision error of the NO$_x$ data (0.2 %). Only at lower concentrations, indeed uncertainties of the NO$_x$ data were higher than 0.2 % of the HONO level. However, caused by the confirmed quadratic dependence (see above), there is no scientific reason, why the NO$_x$ impurities (measured 0.2 % at 20 ppb HONO) should increase again at lower HONO levels. Thus, the statement (>99.8 % at <20 ppb) is correct.

555 ***Page 11, Line 297: The purities cited from these two works are the lower limits, when the sources were challenged to their limits or that of the instrumentation being used to determine impurities. While that is also the case with the value being discussed here, perhaps an additional point to make is that all three of these sources have >98 % purity when operated under ideal conditions in the environmentally relevant range of outputs? This would be a more balanced evaluation.***

First, the purity of our source is >99.8 % at <20 ppb HONO as explained above. In addition, the lower purity published in the
560 other studies is not only caused by the NO$_x$ impurities (see our Fig. 4), but also by impurities of ClNO and HCl (see e.g. the increasing HCl in Fig. 2 of Loa et al. (2020), when the HONO concentration was increased). These impurities are absent in our source by definition. And finally, to the mentioned potential lower limits of the purity by analytical limitation in former studies: By the reasons explained above, a HONO source should not be characterized by any instrument with too low sensitivity, limiting the main conclusions (i.e. purity of the source). The HONO concentration should be increased in such
565 studies unless the error of the used instrument allows a reasonable quantification of the purity. If the purity of >98% claimed

by the referee should be confirmed by experimental data, e.g. in the study of Loa et al. (2020), at least 5 times higher HONO levels have to be used. As long as this is not done, we can only refer to the lower limit purity given in the cited papers, which we have done correctly in that sentence.

*Page 11, Lines 300-301: Give quantitative advantages. These do not seem particularly specific. The Authors point out limitations in the discussion that impedes some of these statements (e.g. reagent consumption rate can be a major drawback). Also, for instruments like a ToF-CIMS utilizing CH3I reagent ion chemistry, the RH variance in the calibration flow would make the calibrations more difficult due to the water-dependence of the ionisation scheme.*

The quantitative variabilities of the parameters are described in the different sections of the manuscript (i.e. gas flow 0.5-2 L min$^1$, liquid flow rate 0.2-0.8 mL min$^{-1}$ for each reagent, dewpoint: 5-18 °C). While the variation of the gas and liquid flow rates may be indeed not too important (changing only time response and resource consumption), the variability of the temperature/dewpoint is an important parameter. Here the referee gave an excellent example confirming our statements above. In our source, humidity is absolutely constant, if the temperature of stripping coil is not changed. Under constant humidity, we still can produce variable HONO levels in short time, in contrast to the dilution of a constant HONO source by synthetic air. But still we can change the humidity when requested, e.g. to quantify the non-linear sensitivity of a CIMS instrument with humidity. This was done, for example, for the CIMS used in Jurkat et al., (2011), for which a three-dimensional non-linear calibration was necessary (sensitivity was a function of the HONO level and the humidity). Although this was a huge effort (took several days for calibration: different concentration dependencies were studied, each at a different humidity), this was possible with our HONO source, but not with a source mentioned by the referee, producing only a fixed HONO concentration.

*Page 11, Lines 302-304: This is by far the biggest contribution of this source to the field. Should be the first point.*

In a "dialectic" listing, the last point is always the most important ("take home message at the very end…"). So we would like to leave the order.

*Page 11, Lines 330-331: Yet again, atmospheric mixing ratios of HONO are typically below 5 ppbv. This systematic evaluation of the system by exclusion of the atmospherically-relevant mixing ratio range is not reporting the true performance of the system. Please revise throughout the manuscript to provide performance metrics for the data collected at only 5 ppbv and lower.*

See our answers above.

**References not listed in the manuscript:**

Becker, K. H., Kleffmann, J., Kurtenbach, R. and Wiesen, P., Febo, A., Gherardi, M., and Sparapani, R.: Line Strength Measurements of *trans*-HONO near 1255 cm$^{-1}$ by Tunable Diode Laser Spectrometry, Geophys. Res. Lett., 22, 2485-2488, 1995.

Elshorbany, Y. F., Kleffmann, J., Kurtenbach, R., Lissi, E., Rubio, M., Villena, G., Gramsch, E., Rickard, A. R., Pilling, M. J., and Wiesen, P.: Seasonal dependence of the oxidation capacity of the city of Santiago de Chile, Atmos. Environ., 44, 5383-5394, 2010.

Stutz, J., E. S. Kim, U. Platt, P. Bruno, C. Perrino, A. Febo: UV-Visible Absorption Cross Sections of Nitrous Acid, J. Geophys. Res., 105, 14585-14592, 2000.